# Identification of recurrent dynamics in distributed neural populations

Rodrigo Osuna-Orozco[1]*, Edward Castillo[1], Kameron Decker Harris[2], Samantha R. Santacruz[1]

**1** Department of Biomedical Engineering, University of Texas at Austin, Austin, Texas, United States of America, **2** Department of Computer Science, Western Washington University, Bellingham, Washington, United States of America

* r.osuna.orozco@utexas.edu

## Abstract

Large-scale recordings of neural activity over broad anatomical areas with high spatial and temporal resolution are increasingly common in modern experimental neuroscience. Recently, recurrent switching dynamical systems have been used to tackle the scale and complexity of these data. However, an important challenge remains in providing insights into the existence and structure of recurrent linear dynamics in neural time series data. Here we test a scalable approach to time-varying autoregression with low-rank tensors to recover the recurrent dynamics in stochastic neural mass models with multiple stable attractors. We demonstrate that the parsimonious representation of time-varying system matrices in terms of temporal modes can recover the attractor structure of simple systems via clustering. We then consider simulations based on a human brain connectivity matrix in high and low global connection strength regimes, and reveal the hierarchical clustering structure of the dynamics. Finally, we explain the impact of the forecast time delay on the estimation of the underlying rank and temporal variability of the time series dynamics. This study illustrates that prediction error minimization is not sufficient to recover meaningful dynamic structure and that it is crucial to account for the three key timescales arising from dynamics, noise processes, and attractor switching.

**Data Availability Statement:** All code written in support of this publication is publicly available at https://github.com/rosunaor/recurrent_dynamics.

## Author summary

How can we make sense of the complex data we obtain from brain recordings? Linear systems are the most straightforward approach for describing brain dynamics. Many techniques have been developed that use sets of linear systems to predict and understand brain activity, including the discovery of recurrent linear dynamics. In this study, we use simulations of interacting neural populations to test whether linear recurrent dynamics are an appropriate description for noisy brain activity. Using a data-driven model, we uncover the neural dynamic structure and delineate the conditions under which there exist separate well-defined linear regimes. Moreover, we reveal the importance of looking beyond prediction error minimization in fitting linear models. We arrive at the seemingly paradoxical conclusion that accounting for the contributions of noise and regime

**Funding:** The author(s) received no specific funding for this work.

**Competing interests:** The authors have declared that no competing interests exist.

switching is easier when predicting farther into the future. The insights and strategies we present will be helpful in making sense of complex brain data while minimizing bias in their interpretation.

## Introduction

With the advent of large-scale recordings of neural populations over broad anatomical areas, a pressing need has arisen for mathematical models that can cope with the complexity and scale of the dynamics captured in these rich datasets [1]. Developing efficient analytical tools to gain well-principled insights from such data remains a fundamental challenge for modern neuroscience. This is in no small part due to the high-dimensional, complex, nonlinear, and nonstationary dynamics of the neural activity that they capture [2, 3]. Linear dynamical systems (LDS) are a common approach to model neural time series as observations of underlying dynamic variables which are often assumed to be of lower dimension than the data [2, 3]. Commonly, LDSs are further considered to be time-invariant [2, 4, 5], an assumption that makes them more tractable. Nonetheless, time-invariant models are often too restrictive to deal with the complex nonstationary data observed in most neural recordings. For instance, time-invariant LDSs cannot capture multistable dynamics as observed in resting state [6] or as hypothesized to underlie decision-making [7]. Time-invariant LDSs also cannot account for the possibility of chaotic behavior [8] or describe varying spectral power commonly observed in neural field data [9, 10].

A natural candidate to overcome some of these limitations is a time-varying model. Such a model allows for a unique LDS at each time point. However, this comes at the cost of vastly increased complexity as well as reduced interpretability and usability for prediction of unseen data. To leverage benefits between these two time-invariant and time-varying alternatives, recurrent switching LDSs allow for a prescribed number of linear systems that describe the evolution of the system at different times [10–14]. However, unsupervised determination of both the existence of switching dynamics as well as of the appropriate number of subsystems remains challenging. Most existing approaches assume a fixed number of underlying dynamical systems to describe observed data and hence do not directly address the issue of how many underlying dynamical systems are appropriate for a given data set [11, 15, 16]. Alternatively, likelihood-based detection of dynamical breaks can be costly for large-scale neural data with rich temporal structure [12].

To address this problem, we leverage a highly scalable method for time-varying autoregression named Time-Varying Autoregression with Low-Rank Tensors (TVART) [9]. TVART separates a multivariate time series into non-overlapping windows and considers a separate affine model parametrized by a system matrix and a constant vector. By stacking the matrices into a tensor and enforcing a low-rank constraint, TVART easily scales to high-dimensional data. Moreover, this method provides a low-dimensional representation of the dynamics by using the canonical polyadic decomposition [17]. The low-dimensional representation can then be leveraged for clustering the dynamical system matrices.

Our previous work has demonstrated the use of TVART for the characterization of nonstationary dynamics in simulations and non-human primate electrophysiological data [10]. Nevertheless, a major gap remains in establishing the validity of switching linear dynamical systems models as meaningful representations of neural activity. It is well known that neural activity, from single neurons to networks of brain areas, is inherently non-linear [18, 19]. Here, we aim to establish whether identifying time-varying dynamics results in linear systems

with a meaningful correspondence to the properties of an underlying non-linear system. We are interested in whether the identified linear systems correspond to attractors. Attractors of neural dynamics have been posited to correspond to important qualitative states such as memory representations and decisions [7, 20]. Concretely, we test the sensitivity of this identification approach to increasing stochastic forcing and increasing complexity of the attractor landscape since significant stochasticity and complex high-dimensional dynamics are commonplace for large-scale neural data. Thus, by demonstrating that TVART recovers the attractor structure and timing of switching between attractors, we establish it as a robust basis for switching linear dynamical systems representations of neural activity. Moreover, TVART also allows us to determine whether or not switching linear systems are an appropriate approximation to observed neural dynamics.

In this study, we explore in detail the capabilities of TVART as an unsupervised and parsimonious (given the low-dimensional representation of the fitted linear systems) approach to identifying different dynamical neural regimes. To this end, we focus on simulations of neural mass models that exhibit multiple stable fixed-point attractors. Such models have been shown to be a biophysically sound and accurate description of human whole brain activity [6, 18, 21]. We focus on two main cases where these neural mass models have been applied to neuroscience problems. The first entails two interacting neural populations whose activity can be used to model decision-making [7]. The second case is based on whole-brain human structural connectivity where the neural mass model can be used to study functional connectivity dynamics [21, 22]. For both cases, we explore simulations with varying numbers of attractors and at increasing noise levels. Increasing the value of either of these variables makes identifying well-separated switching linear dynamical systems a more dubious endeavor.

Moreover, stochastic forcing is essential for neural mass models to account for the variability inherent to populations of spiking neurons [7]. It also influences the probability distribution of neural states, the frequency of switching across attractors, and how far from equilibrium the system is on average. In the context of decision-making, more attractors would correspond to more possible choices, while more stochasticity could correspond to higher uncertainty. Here, we demonstrate that clustering based on the low-dimensional representation of neural activity obtained by TVART can extract the exact number and timing of different attractor dynamics even in the presence of strong stochastic forcing. TVART is also used to gain insights regarding the temporal variability of the dynamics of models based on human brain connectivity matrices for connection strengths that lead to different attractor landscapes. Whole-brain dynamics can potentially sustain a vast number of attractors with ensuing dynamics that are very far from equilibrium. For such a scenario, it is vital to determine whether a recurrent linear systems approach faithfully captures the dynamics or if including the nonlinearity is essential. The results presented here illustrate how TVART produces insights that go beyond offering an accurate forecast model.

Furthermore, we explore the role of prediction delay in both prediction error and attractor recovery. We demonstrate that the prediction error scales exponentially with the prediction delay, with exponents given by the eigenvalues of the underlying stable fixed points. Through our simulations, we show that there appears to be an optimal prediction delay for the identification of different attractors in noisy settings. We show how increasing stochastic forcing degrades the capability to identify attractors for a given prediction horizon. Our results suggest that approximating neural time series with a set of linear dynamical systems can greatly benefit by optimizing prediction delay to balance error and dynamical regime identification.

In the following sections, we first describe the neural mass model used for our simulations and show the possible multistable regimes it can support. Then we describe how optimizing the prediction horizon helps tackle increasing dynamic complexity (e.g. increasing the number

of attractors) and stochastic noise for attractor recovery. We then demonstrate how prediction error converges exponentially with prediction delay and analytically derive an expression that explains this convergence. Subsequently, we show how identifying temporal variability of dynamics also necessitates optimization of prediction delay and illustrate when increased temporal resolution can overcome rank penalties (which make for simpler models). We also describe the clustering of high-dimensional multistable dynamics via TVART. Finally, we discuss our results and their relevance for data-driven explorations of brain dynamics.

## Results

### Neural mass model

Neural mass models have been used to explain whole brain observations, with their capacity for multistability offering insights into resting state functional connectivity dynamics [6, 18, 22]. We investigate the multistable dynamics of interacting populations of neurons using a biophysical neural mass model [6, 7, 21]. Here, the term population refers to groups of neurons whose activity can be approximated by a collective firing rate [7]. This model consists of the following system of stochastic differential equations (SDEs):

$$\frac{d\mathbf{S}}{dt} = -\frac{1}{\tau_s}\mathbf{S} + \gamma(\mathbf{1} - \mathbf{S}) \odot \mathbf{R} + \sigma\boldsymbol{\eta}(t), \tag{1}$$

$$R_i = \frac{ax_i - b}{1 - \exp[-d(ax_i - b)]}, \tag{2}$$

$$\mathbf{x} = wJ_N\mathbf{S} + GJ_N\mathbf{CS} + \mathbf{I_0}, \tag{3}$$

where $\mathbf{S}$ and $\mathbf{R}$ are the average synaptic gating (unitless) and firing rates (Hz) for all populations, respectively; $\mathbf{x}$ represents the total synaptic input for a population; $\boldsymbol{\eta} = d\mathbf{W}/dt$ is the derivative of the Wiener process, $\mathbf{W}$, and the noise amplitude is given by the scalar $\sigma$ [23]. Here we use the symbol $\odot$ to denote the Hadamard product (element-wise multiplication). The neural populations interact via the structural connectivity matrix $C$, and the strength of the coupling can be modulated via the parameter $G$. Complete parameter values are provided in Table 1 (see Methods).

To test the proposed approach to identifying recurrent dynamics, we first simulate models of two interacting neural populations. We adjust parameters $w$, $J_N$, and $G$, so that the attractor landscape varies in complexity (see Table 1 in Methods). In particular, we consider a case with

**Table 1. Parameter descriptions and values used for MFM simulations.**

| Constant | Description | Values for 2 populations | Values for 90 populations |
|---|---|---|---|
| $\tau_s$ | Decay time constant | 0.1 s | 0.1 s |
| $\gamma$ | Kinetic parameter | 0.641 | 0.641 |
| $a$ | Sigmoid parameter | 270 1/nC | 270 1/nC |
| $b$ | Sigmoid parameter | 108 Hz | 108 Hz |
| $d$ | Sigmoid parameter | 0.154 s | 0.154 s |
| $w$ | Local excitatory recurrence | 0.9, 1.07 | 0.95 |
| $J_N$ | Synaptic Coupling | 0.2609, 0.29 nA | 0.2609 nA |
| $G$ | Coupling strength | 0.32, 0.03 | 0.08, 0.55 |
| $I_0$ | Overall effective external input | 0.3 nA | 0.32 nA |
| $\sigma$ | Noise amplitude | 0.3-1 | 0.3-1 |

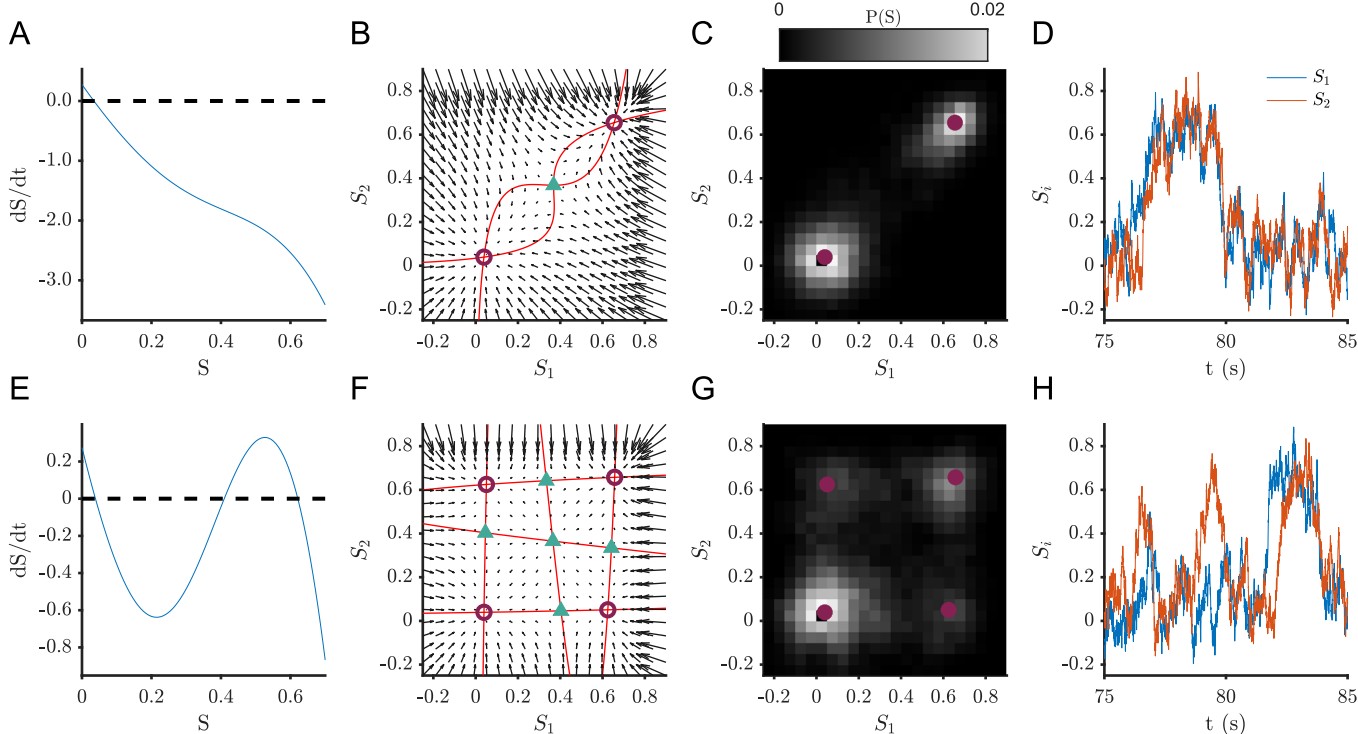

**Fig 1. Dynamics of two representative population models exhibiting multistability.** Plots in top row correspond to 2 fixed-point attractor case and plots in bottom row correspond to 4 fixed-point attractor case. Single population ($G = 0$) derivative for parameters yielding (A) two and (E) four fixed-point attractors. (B, F) Phase portrait showing nullclines in red, fixed-point attractors as circles and unstable fixed points as triangles. (C, G) Probability distribution in phase space (fixed-point attractors shown as blue circles) for noise level $\sigma = 0.4$ Hz (D, H) Example time series data corresponding to the probability distributions in C and G.

two stable fixed-point attractors (Fig 1A–1D) and a case with four stable fixed-point attractors (Fig 1E–1H). The case with two fixed-point attractors was originally proposed by Wong and Wang to explain binary choice decision-making [7].

We then consider simulations based on the structural connectivity matrix of a healthy human subject derived from MR images [24] comprised of 90 regions of interest (Fig 2A). To explore the attractor landscape prescribed by the structural connectivity matrix, we fix the values of all parameters except for the coupling strength $G$. The values of the other parameters are similar to previous work where the mean field model (Eq (1)) was used to reproduce functional connectivity dynamics [22].

For the chosen parameters, the number of fixed point attractors, their average activity, and their variability, vary quite substantially as a function of the coupling strength (Fig 2B and 2C). Notably, mean fixed point activity increases monotonically with the coupling strength (Fig 2B). At the lowest coupling strength, there is a single fixed point with low average activity, and all neural populations, being effectively decoupled and having the same parameters, have the same level of activity. At the highest coupling strength, there is a single fixed point but different populations have different levels of activity, so that the overall variance is nonzero (Fig 2B and 2C).

Moreover, for the regime where there is multistability, there is a transition in the neural populations that exhibit high variance. Namely, at low coupling strength, it is the populations with strong total incoming connections that have high variance across their stable fixed points.

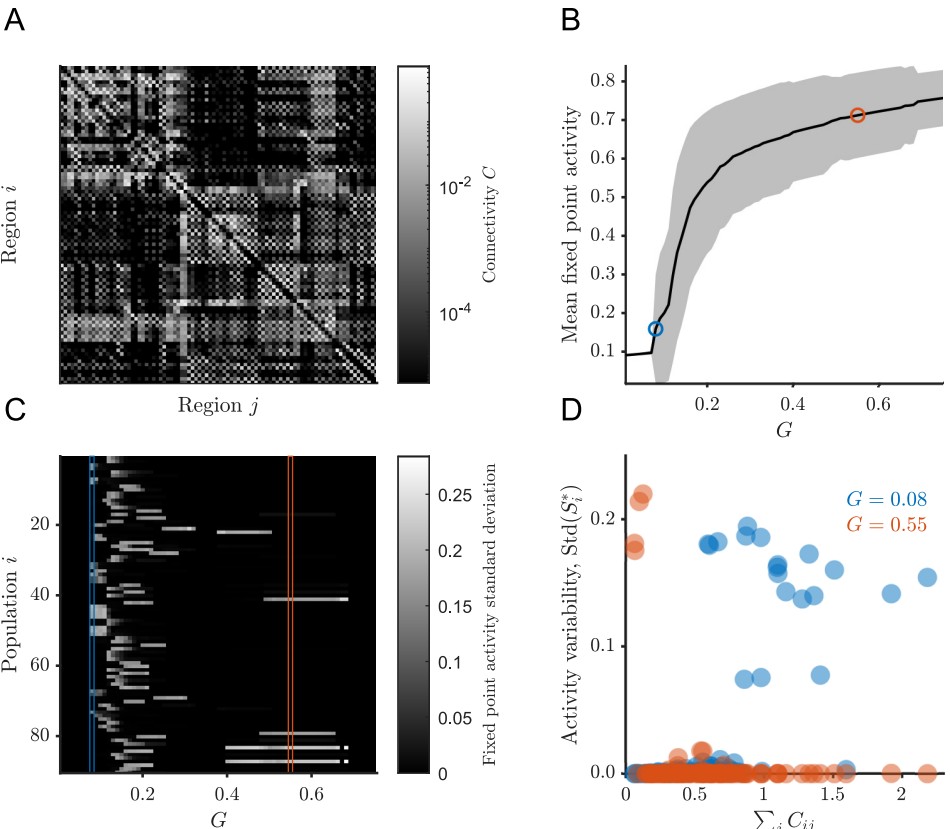

**Fig 2. Attractor landscape of brain connectivity-based simulations varies with coupling strength.** (A) Structural connectivity matrix from a healthy human subject obtained from Skoch et al. [24]. (B) The average activity over the stable fixed points at different values of $G$, shaded region corresponds to two standard deviations. (C) Standard deviation of activity over the stable fixed points at different values of $G$. (D) Standard deviation of the fixed point activity, $S_i^*$, calculated over all attractors ordered by incoming connection strengths. The stochastic simulations considered here correspond to the values and columns enclosed in red and blue in (B) and (C), respectively.

In contrast, at high coupling strength, it is the neural populations with weak total incoming connections that exhibit high variance across their stable fixed points (Fig 2D). For this study, we considered in detail the values $G = 0.08$ and $G = 0.55$. We considered these two cases, as both exhibit multistability, but the first one does so in a low mean activity regime while the second one does so in a high mean activity regime (Fig 2B–2D). The choice of these values is discretionary and similar comparisons would hold for simulations in comparable regimes.

## Attractor recovery benefits from an increased prediction horizon

TVART estimates affine models for non-overlapping windows [9], describing the evolution of the system according to:

$$\mathbf{x}(t + \Delta t) = \mathbf{A}_k \mathbf{x}(t) + \mathbf{b}_k + \text{Error}, \tag{4}$$

where there is a unique system matrix, $\mathbf{A}_k$, and offset vector, $\mathbf{b}_k$, for every time window $k$. The prediction delay, $\Delta t$, is typically given by the sampling frequency in a given data set. Both the system matrix and the offset vector depend on the prediction delay even for the same data set. In the case of neural dynamics that exhibit multiple stable states, we aim at recovering system matrices that faithfully reflect this attractor landscape.

In TVART the system matrix fit to window $k$ is expressed as the product of three matrices as follows:

$$\mathbf{A}_k = \mathbf{U}^{(1)}\mathbf{D}^{(k)}\mathbf{U}^{(2)^T}, \tag{5}$$

where $\mathbf{D}^{(k)} = \mathrm{diag}(u_k^{(3)})$, and $u_k^{(3)}$ is the $k^{th}$ row of the temporal mode matrix $U^{(3)} \in \mathbb{R}^{T \times R}$; and $U^{(1)}, U^{(2)} \in \mathbb{R}^{N \times R}$ are called the left and right spatial modes, $R$ is the imposed rank constraint, and the time series is split into $T$ windows of equal duration $M$. The spatial modes are constant for the whole time series [9]. We can identify switching dynamics by examining the temporal modes, $u_k^{(3)}$. Specifically, we look for clusters of TVART temporal modes, which indicate temporal windows with similar linear dynamics. Such similar linear dynamics may occur within segments of the time series that exist within the same attractor. Therefore, TVART can facilitate the identification of attractor dynamics.

We first consider simulations for a choice of parameters that results in two attracting fixed points (Fig 3A). TVART was able to recover the correct attractor with high accuracy even for simulations subject to high noise amplitude. We illustrate this point with results for simulations with a high noise amplitude of $\sigma = 0.5$ (Fig 3). For this noise amplitude, the probability density of the dynamic variables has two well-separated peaks around the two stable fixed points (Fig 3A), and the time series shows easily identifiable transitions between the two attractors (Fig 3B).

For these two population simulations, it is possible to depict the temporal modes since they are two-dimensional vectors. We illustrate the clustering of the temporal mode vectors for a

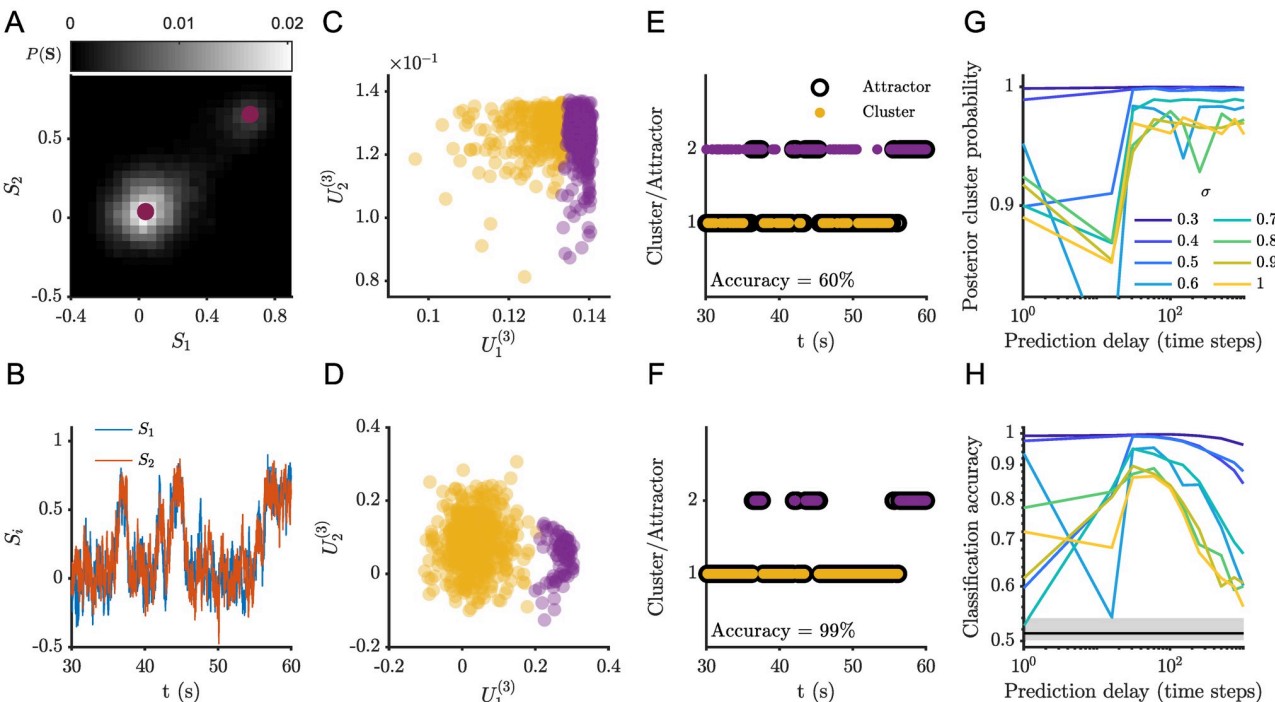

**Fig 3. TVART can recover attractor dynamics with high accuracy for a two attractor network.** (A) Sample simulation ($\sigma = 0.5$) phase space probability distribution. (B) Sample of simulation time series. (C) Clustering of the temporal modes for one step prediction regression and (D) for 31 steps ahead prediction regression. (E) Comparison of the attractor and cluster indices for one step prediction regression and (F) for 31 steps ahead prediction regression, the classification accuracy for (E) was 59.7% and for (F) 99.0%. (G) Mean posterior cluster probability as a function of prediction delay for several noise levels. (H) Classification accuracy as a function of prediction delay for several noise levels, median chance level is indicated by the black horizontal line and the shaded region encompasses the 5th and 95th percentiles.

few simulations for short (Fig 3C) and long (Fig 3D) prediction delays. For these examples, we can see that increasing the temporal delay prediction improves the identification of the attractors. This is apparent from the separation between clusters (Fig 3C and 3D), as well as from the correspondence between cluster identification and underlying attractor over the time series windows (Fig 3E and 3F).

To measure how accurately identified clusters capture attractor hopping, we index each time series window by their proximity to fixed point attractors. Thus, the attractor index represents the attractor the trajectory is closest to on average for a given window. The prediction accuracy quantifies the match rates between the attractor and cluster indices. (Fig 3E and 3F). We matched temporal mode cluster indices to attractor indices by choosing the permutation that minimizes the difference between the two. We also quantify the mean posterior cluster probability as the mean over all observations of each observation's maximum posterior probability, so that with perfect clustering we obtain a mean posterior cluster probability of one (Fig 3G and 3H). In all cases both the posterior cluster probability and the classification accuracy decrease with increasing noise amplitude (Fig 3G and 3H). Nonetheless, for all noise amplitudes the classification accuracy remains above chance level (50%) and is near 100% under low noise conditions.

Interestingly, both the mean posterior cluster probability and the classification accuracy show maxima as functions of prediction delay (Fig 3G and 3H). Initially, increasing the time delay of the prediction resulted in bigger differences among the fitted system matrices. Thus, increasing prediction delay results in clusters of temporal mode vectors that are more clearly separated, allowing better classification accuracy (Fig 3H). But as the prediction delay grows too large, switching between the attractors impacts the results. We computed the average time for attractor hopping and it has a strong dependence on the stochastic forcing. The mean time for transitions between different attractors spans between 2700 steps for the lowest stochastic forcing in the two attractor case and 19 steps for the highest stochastic forcing in the four attractor case. This means that for long enough delays the fitted model would try to predict observations between different attractors, with dynamics that correspond to neither.

To explore the impact of a more complex attractor landscape, we perform simulations involving four fixed point attractors (Fig 4). For the same noise level, $\sigma = 0.5$, the four attractor case does not show very distinct peaks in the probability density function (Fig 4A) and the transitions between the different attractors are harder to detect in the time series (Fig 4B). Nonetheless, increasing the prediction delay yielded similar trends with respect to temporal mode clustering and attractor identification. As in the two attractor case, increasing prediction delay results in better separation between temporal mode clusters (Fig 4C and 4D). Similarly, increasing the prediction delay improved the classification accuracy (Fig 4E and 4F).

Naturally, as the number of attractors increases, accurate classification becomes more challenging. We indeed observe lower classification accuracy in this case (Fig 4H), although performance is always above chance level (25%). In contrast to the two attractor case, for the four-attractor system, only the classification accuracy shows a clear maxima as a function of the prediction delay (Fig 4G and 4H).

It is worth emphasizing that the maxima of the classification accuracy always occur at values that are suboptimal in terms of prediction error (as computed by the mean squared error for the autoregressive model prediction and the observed time series values). As is expected from stochastic simulations, fitting models to the shortest time delays yields the lowest prediction errors. In the following sections, we illustrate how TVART produces predictions that are consistent with errors due to linearization and inherent stochastic forcing.

We also observed a similar effect on the classification of dynamics using an alternative approach. Decomposed linear dynamical systems (dLDS) [16] describe dynamics using

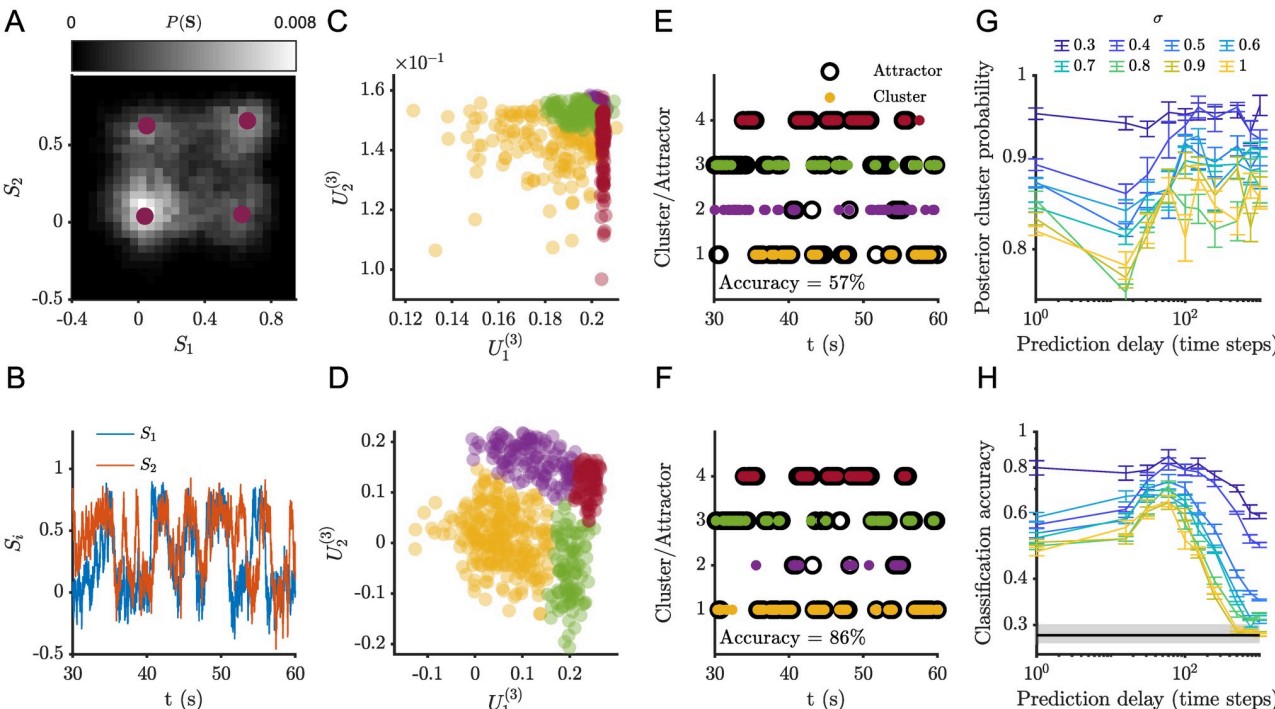

**Fig 4. There is an optimal prediction delay for identification of attractor dynamics.** (A) Sample simulation ($\sigma = 0.5$) phase space probability distribution. (B) Sample simulation time series (just a subset is depicted for clarity). (C) Clustering of the temporal modes for one step prediction regression and (D) for 31 steps ahead prediction regression. (E) Comparison of the attractor and cluster indices for one step prediction regression and (F) for 31 steps ahead prediction regression, the classification accuracy for (E) was 56.8% and for (F) 86.1%. Mean posterior cluster (G) Mean posterior cluster probability as a function of prediction delay for several noise levels. (H) Classification accuracy as a function of prediction delay for several noise levels. For (G) and (H) results are averaged over 10 TVART fits and error bars represent the standard error of the mean).median chance level is indicated by the black horizontal line and the shaded region encompasses the 5th and 95th percentiles.

coefficients for a dictionary of linear systems. So, the system matrix is a linear combination of dictionary matrices, with coefficients specified at each time step. We fit the simulations explored in Fig 3 (2 populations, 2 equilibria simulations) and found that classification accuracy can also be improved with longer prediction delays (Fig 5A–5C). Moreover, taking the absolute value of the dynamics coefficients and smoothing them (moving average centered boxcar filter over 100 steps) in time substantially improved the classification accuracy for attractor identification (Fig 5D–5F). Similarly, we replicated the results in Fig 4 (2 populations, 4 equilibria simulations) using dLDS with four dictionary elements. In the case of the four attractor simulations, clustering based on dLDS coefficients only fared slightly above chance levels in identifying attractor occurrence (Fig 6A–6C). In this case, taking the absolute value and smoothing the coefficients markedly increased classification accuracy (Fig 6A–6C).

## Prediction error converges exponentially with prediction delay

The results in the subsection above indicate that looking at prediction delays longer than the sampling period may be beneficial for identification of the attractor dynamics. Since time-varying autoregressive models describe continuous dynamics by discrete approximations, they are sensitive to the delay between the observations that define the regression. For our TVART models, the prediction error for the full rank autoregressive models is observed to increase with increasing prediction delay, with a scaling behavior of the mean square error (MSE) that

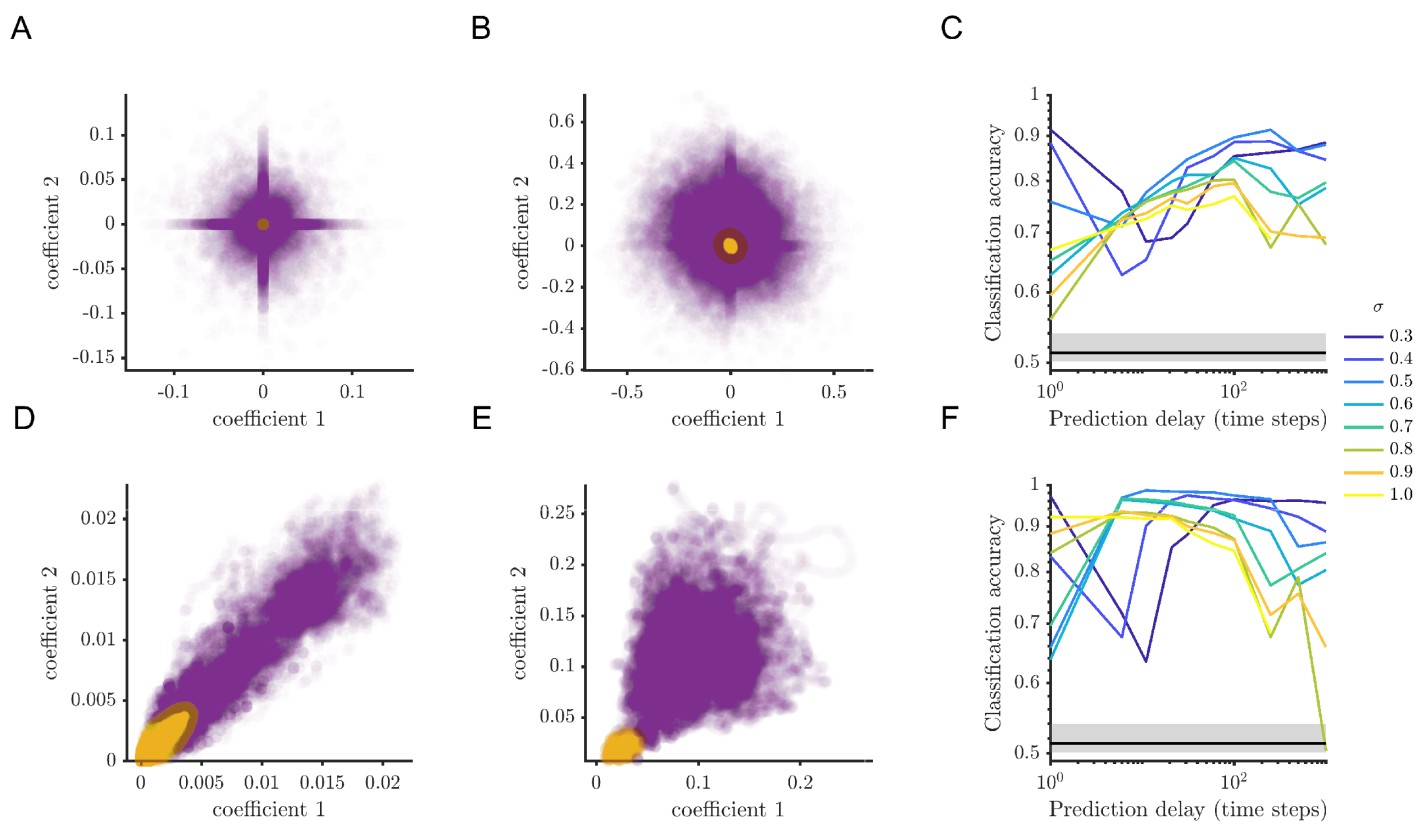

**Fig 5. dLDS attractor recovery also benefits from increasing prediction delay.** dLDS coefficients for the sample simulation with ($\sigma = 0.5$). (A) Coefficients for 1-step delay fits and (B) for 31-step delay fits. (C) Classification accuracy based on dLDS coefficients as a function of prediction delay. (D) dLDS coefficients after taking absolute value and smoothing by a centered boxcar filter over 100 steps for 1-step delay fits and for (E) 31-step delay fits. (F) Classification accuracy based on smoothed absolute value dLDS coefficients as a function of prediction delay. Transforming the dLDS coefficients substantially improved attractor identification. The classification accuracy corresponding to the scatter plots are: (A) 71.4%, (B) 81.7%, (D) 58.1%, and (E) 98.2%.

can be approximated by:

$$\text{MSE} \approx -\frac{\sigma^2}{2\bar{\lambda}}(1 - \exp(2\bar{\lambda}\Delta t)), \tag{6}$$

where $\bar{\lambda}$ is the mean eigenvalue for the Jacobians of all the fixed points. This expression holds not only for the low-dimensional simulations (Fig 7A and 7B), but also for simulations based on the human brain structural connectivity (Fig 7C and 7D). Although human brain simulations have a large number of eigenvalues and attractors, and the dynamics are often far from the fixed points of the system, this approximation method (Eq 6) still faithfully captures the error scaling. This result is robust to variations in the fitting window width and choice of regularization method, spline or total variation, for the chosen regularization strengths (Fig 7C and 7D) [9]. Moreover, we observe that the errors estimated by TVART fall close to the curves of the form of Eq 6, but with the mean eigenvalue replaced by the minimum and maximum eigenvalues (solid black lines, Fig 7).

For the two population simulations, the scaling in Eq 6 holds independently of the number of attractors or the stochastic noise level (Fig 7A and 7B). For simulations based on the human brain connectivity matrix, we tested how well TVART fits with different window sizes. Although the error increases with decreasing window size, the error remains within the

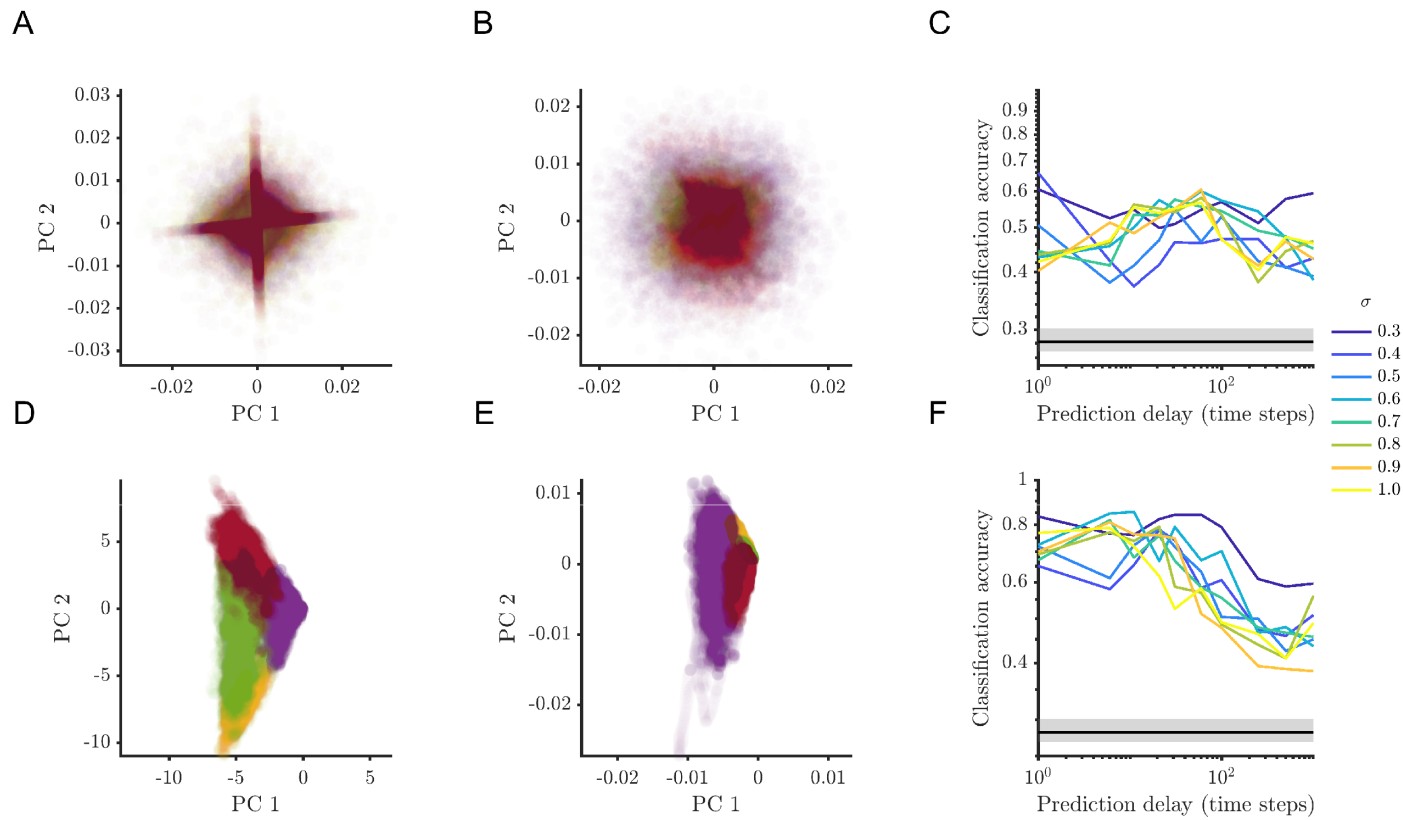

**Fig 6. dLDS attractor recovery benefits from temporal smoothing.** dLDS coefficients for the sample simulation with ($\sigma = 0.5$). (A) Principal components of coefficients for 1-step delay fits and (B) for 31-step delay fits. (C) Classification accuracy based on dLDS coefficients as a function of prediction delay. (D) Principal components of dLDS coefficients after taking absolute value and smoothing over 100 steps for 1-step delay fits and for (E) 31-step delay fits. (F) Classification accuracy based on smoothed absolute value dLDS coefficients as a function of prediction delay. Transforming the dLDS coefficients substantially improved attractor identification. The classification accuracy corresponding to the scatter plots are: (A) 47.9%, (B) 40.7%, (D) 68.6%, and (E) 76.8%.

bounds suggested by Eq 6 for the prediction delays studied. Similarly, increasing the weight of the regularization term increases the MSE while preserving the scaling behavior. These results suggest that TVART approximates the true dynamics as the prediction error from the fitted models are bound by the intrinsic stochastic variation of the dynamics.

We also validated the error scaling using alternative approaches SLDS [11], rSLDS [13], and dLDS [16]. We see that in many cases the errors scale according to the exponential trend described by Eq (6) as seen by curve fitting (Fig 8). In the case of TVART, the errors follow Eq (6) remarkably well with $R^2$ values of 0.95 (Fig 8A), 0.97 (Fig 8B), 0.99 (Fig 8C), and 0.95 (Fig 8D). For SLDS, the errors follow Eq (6) well only for the two population simulations with $R^2$ values of 0.77 (Fig 8A) and 0.96 (Fig 8B). Similarly for rSLDS, the errors follow Eq (6) well only for the two population simulations with $R^2$ values of 0.88 (Fig 8A) and 0.90 (Fig 8B). Both SLDS and rSLDS had mean squared errors much larger than those described by Eq (6) for simulations based on human connectivity with ninety neural populations (Fig 8C and 8D). This was the opposite for dLDS where Eq (6) only described well the results for the 90 population simulations with $R^2$ values of 0.997 (Fig 8C), and 0.99 (Fig 8D). In the case of dLDS, the errors where much lower than those described by Eq (6), which suggests that dLDS is able to recover a more accurate linearization of the nonlinear dynamics away from fixed points.

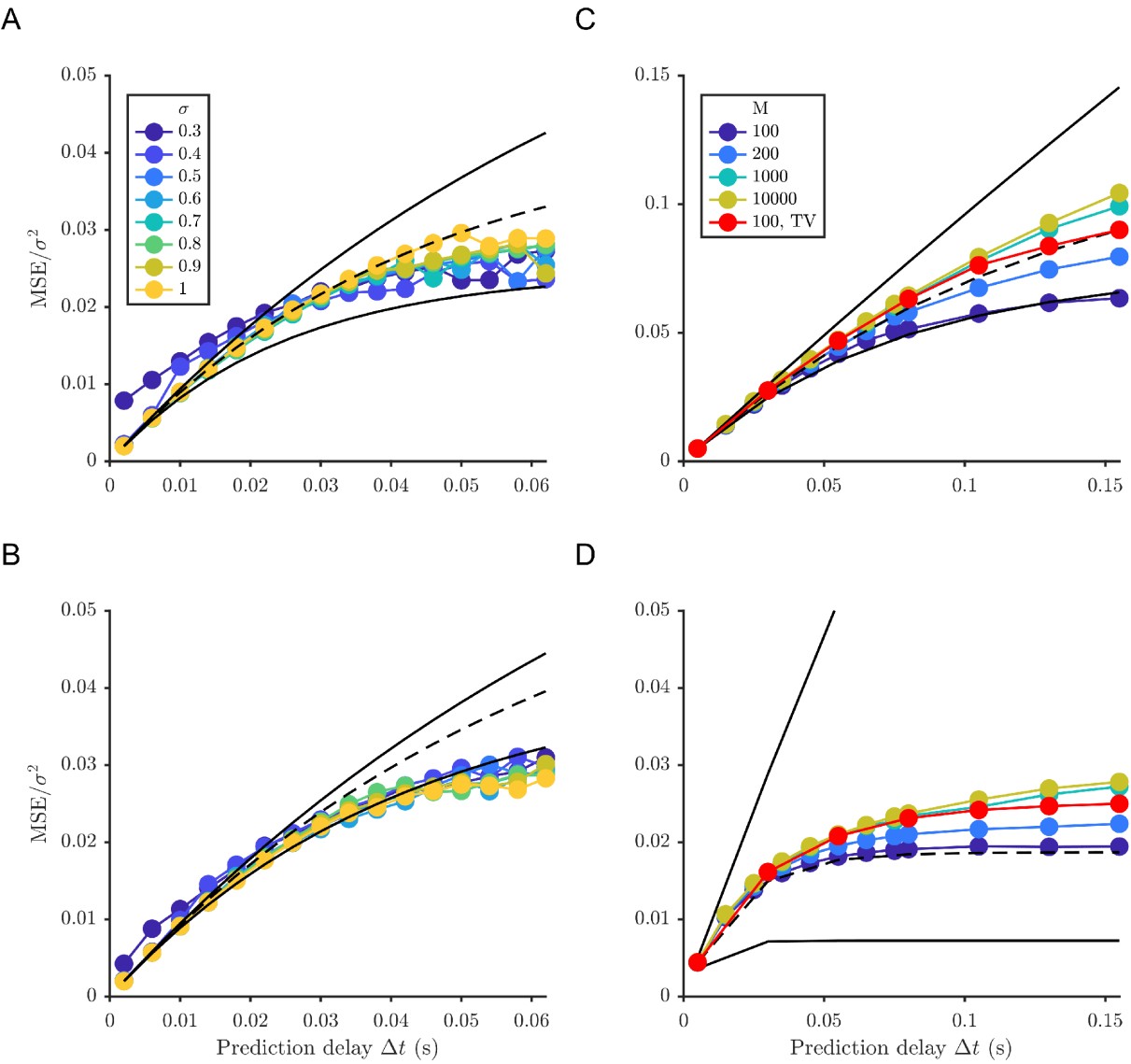

**Fig 7. Exponential scaling of prediction error as a function of prediction delay.** Normalized mean square error as a function of prediction delay $\Delta t$ compared to Eq (4) for the mean (dashed black), maximum (solid black) and minimum (solid black) eigenvalues of the Jacobians for all the fixed points. Results for: (A) 2 populations, 2 equilibria simulations; (B) 2 populations, 4 equilibria simulations; (C) 90 populations, 13 equilibria simulations (G = 0.08); (D) 90 populations, 8 equilibria simulations (G = 0.55). Results in red are for a total variation and the rest are for spline regularization penalties.

## Derivation of MSE around a single stable fixed point

To make sense of the sources of error, we now analytically derive an expression similar to Eq (6). Consider the nonlinear system of stochastic differential equations:

$$\frac{d\mathbf{x}}{dt} = \mathbf{F}(\mathbf{x}) + \sigma \frac{d\mathbf{W}}{dt}. \tag{7}$$

Linearizing about a fixed point, $\mathbf{x_0}$, $\mathbf{F}(\mathbf{x_0}) = \mathbf{0}$, and letting $\tilde{\mathbf{x}} = \mathbf{x} - \mathbf{x}_0$, we obtain:

$$\frac{d\mathbf{x}}{dt} = \frac{d\tilde{\mathbf{x}}}{dt} \approx \mathbf{J}(\mathbf{x_0})\tilde{\mathbf{x}} + \sigma \frac{d\mathbf{W}}{dt}, \tag{8}$$

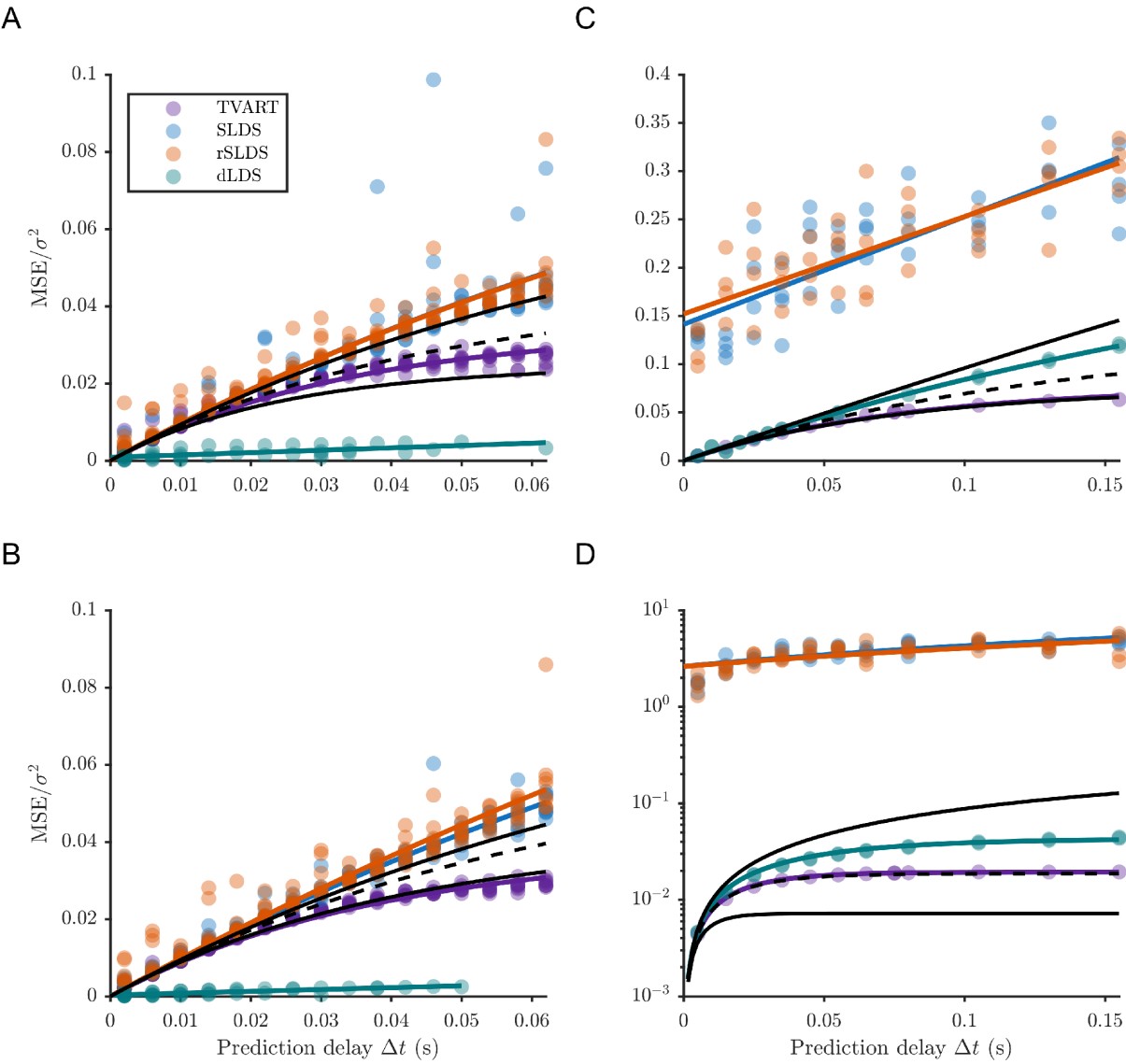

**Fig 8. Exponential scaling of prediction error as a function of prediction delay for several methods.** Normalized mean square error as a function of prediction delay $\Delta t$ compared to Eq (4) for the mean (dashed black), maximum (solid black) and minimum (solid black) eigenvalues of the Jacobians for all the fixed points (derived from Eq (1)). Results for: (A) 2 populations, 2 equilibria simulations; (B) 2 populations, 4 equilibria simulations; (C) 90 populations, 13 equilibria simulations (G = 0.08); (D) 90 populations, 8 equilibria simulations (G = 0.55). Solid lines in color are trend lines based on the best of either a linear fit or a fit following Eq (6).

where $\mathbf{J}$ is the Jacobian $\mathbf{J} = \left[\frac{\partial \mathbf{F}}{\partial x_1}, \dots, \frac{\partial \mathbf{F}}{\partial x_n}\right]$. Eq 8 defines a multivariate Ornstein-Uhlenbeck process. Integrating Eq 8 [23], we get:

$$\tilde{\mathbf{x}}(t) = e^{\mathbf{J}(\mathbf{x_0})t}\tilde{\mathbf{x}}(0) + \sigma \int_0^t e^{\mathbf{J}(\mathbf{x_0})(t-t')}d\mathbf{W}(t'), \tag{9}$$

$$\mathbf{x}(t) = e^{\mathbf{J}(\mathbf{x_0})t}\mathbf{x}(0) + (\mathbf{I} - e^{\mathbf{J}(\mathbf{x_0})t})\mathbf{x_0} + \sigma \int_0^t e^{\mathbf{J}(\mathbf{x_0})(t-t')}d\mathbf{W}(t'). \tag{10}$$

To make the comparison to time-varying linear dynamical systems more transparent, we rewrite Eq 10. We consider a time series where the state at time $t + \Delta t$ is predicted from the observed state at time $t$. In this case, we know our initial condition is $\mathbf{x}(t)$, so by a change of variables $t \rightarrow t + \Delta t$ we can write:

$$\mathbf{x}(t + \Delta t) = e^{\mathbf{J}(\mathbf{x_0})\Delta t}\mathbf{x}(t) + (\mathbf{I} - e^{\mathbf{J}(\mathbf{x_0})\Delta t})\mathbf{x_0} + \sigma \int_0^{\Delta t} e^{\mathbf{J}(\mathbf{x_0})(\Delta t - t')}d\mathbf{W}(t'). \tag{11}$$

We can thus compare the terms in Eq 11 to those in Eq 4. That is, $\mathbf{A}_k \approx e^{\mathbf{J}(\mathbf{x_0})\Delta t}$, $b_k \approx (\mathbf{I} - e^{\mathbf{J}(\mathbf{x_0})\Delta t})\mathbf{x_0}$, and Error $\approx \sigma \int_0^{\delta t} e^{\mathbf{J}(\mathbf{x_0})(\delta t - t')}d\mathbf{W}(t')$.

Upon inspection, Eq 11 is consistent with what we would expect for stochastic dynamics around a stable fixed point; namely, the expected value tends to approximate the fixed point $\mathbf{x_0}$. In the limit as $\Delta t \rightarrow 0$, the system matrix $\exp(\mathbf{J}(\mathbf{x_0})\Delta t) \rightarrow \mathbf{I}$, where $\mathbf{I}$ is the identity matrix. On the other hand, as $\Delta t \rightarrow \infty$, if the eigenvalues of $\mathbf{J}(\mathbf{x_0})$ have negative real part (as they ought to for a fixed point attractor) the system matrix vanishes and the expected value of the prediction is simply the stable fixed point.

To make sense of the prediction error in an autoregressive process, we can look at the conditional variance associated with the random process expressed in Eq 11:

$$\begin{aligned} \text{Var}[\mathbf{x}(t + \Delta t)|\mathbf{x}(t)] &= \left\langle \left[ \int_0^{\Delta t} \sigma e^{\mathbf{J}(\mathbf{x_0})(\Delta t - t')}d\mathbf{W}(t') \right] \left[ \int_0^{\Delta t} \sigma e^{\mathbf{J}(\mathbf{x_0})(\Delta t - t')}d\mathbf{W}(t') \right]^T \right\rangle, \\ &= \sigma^2 \int_0^{\Delta t} e^{\mathbf{J}(\mathbf{x_0})(\Delta t - t')}e^{\mathbf{J}(\mathbf{x_0})^T(\Delta t - t')}dt', \end{aligned} \tag{12}$$

where the angled brackets denote the expectation relative to the noise process and the last equality is given by the Itô Isometry [25].

If we assume the Jacobian is normal so that $\mathbf{J}(\mathbf{x_0})\mathbf{J}(\mathbf{x_0})^T = \mathbf{J}(\mathbf{x_0})^T\mathbf{J}(\mathbf{x_0})$, then there exists a unitary matrix $\mathbf{S}(\mathbf{SS}^T = \mathbf{I})$ such that $\mathbf{SJ}(\mathbf{x_0})\mathbf{S}^T = \Lambda$ and $\mathbf{S}^T\Lambda\mathbf{S} = \mathbf{J}(\mathbf{x_0})$ for the eigenvalue diagonal matrix $\Lambda$. Hence, we obtain:

$$\int_0^{\Delta t} e^{\mathbf{J}(\mathbf{x_0})(\Delta t - t')}e^{\mathbf{J}(\mathbf{x_0})^T(\Delta t - t')}dt' = \mathbf{S}^T \int_0^{\Delta t} e^{\Lambda(\Delta t - t')}e^{\Lambda(\Delta t - t')}dt'\mathbf{S}, \tag{13}$$

$$\text{Var}\left[\mathbf{x_{t+\Delta t}}|\mathbf{x_t}\right] = -\left(\frac{\sigma^2}{2}\right)\mathbf{S}^T\Lambda^{-1}(\mathbf{I} - \exp(2\Lambda\Delta t))\mathbf{S}. \tag{14}$$

This expression can be further related to the predicted MSE by taking the trace:

$$\text{MSE} = -\left(\frac{\sigma^2}{2}\right)\sum_i \frac{1}{\lambda_i}(1 - \exp(2\lambda_i\Delta t)). \tag{15}$$

Although we derived Eq 15 for linearization around a single fixed point, we can compare it to the error scaling observed in our TVART models, Eq 6. For the case of multistable systems, we expect the MSE to be approximated by a linear combination of terms of the form of the right hand side in Eq 15.

## Prediction delay behavior uncovers temporal variability of dynamics

As we consider systems that involve many brain areas, model complexity becomes a concern. Perhaps the two most critical parameters for the TVART model are the choice of window width, $M$ (i.e., the number of observations included in a window), and the maximum rank of

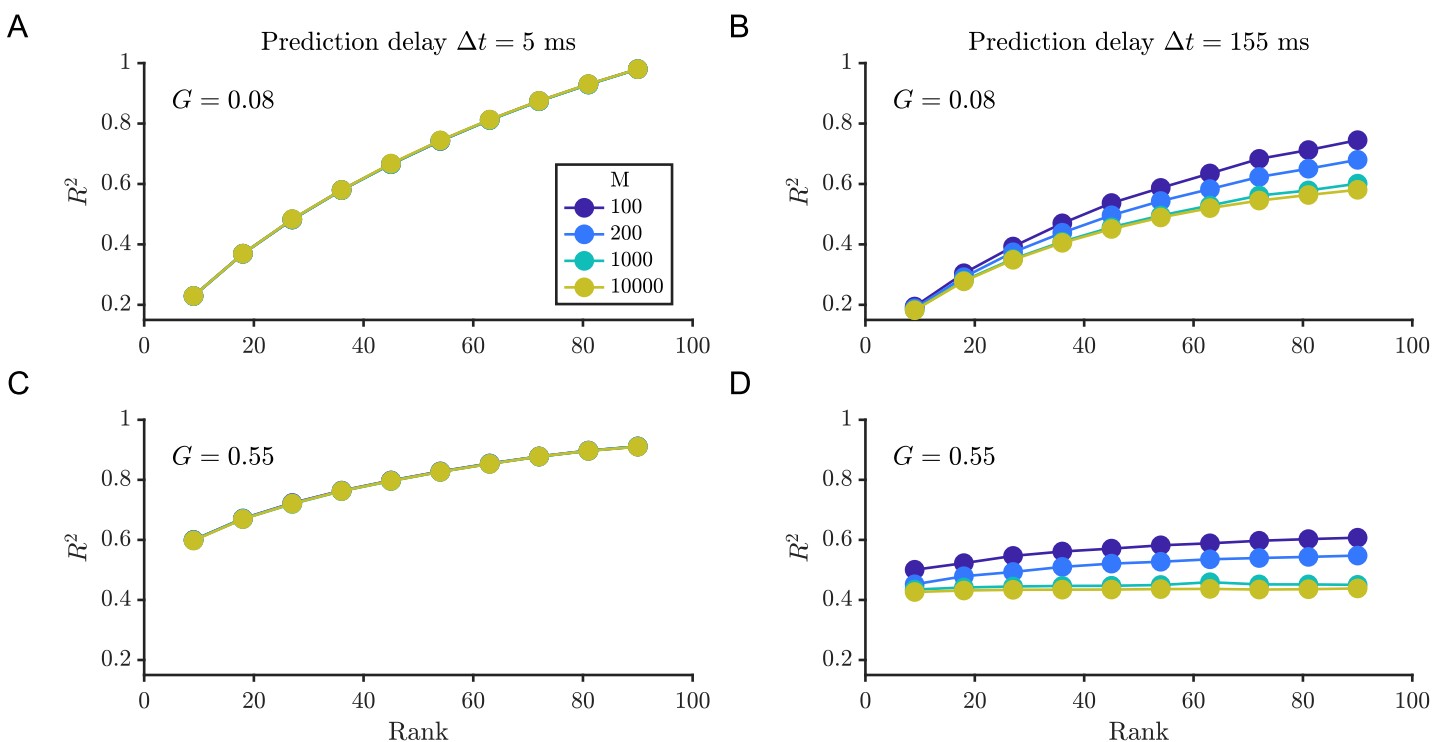

**Fig 9. Identification of temporal variability depends critically on choice or prediction time delay.** $R^2$ values for several different temporal window widths as a function of rank approximation. (A,B) Simulations with G = 0.08; (C,D) simulations with G = 0.55 and for regressions (A,C) based on one step ahead prediction and (B, D) based on 31 step ahead prediction.

the dynamical model. Since both parameters affect model complexity, there is a trade-off between the two when model sparsity is desirable. In this subsection, we consider results for simulations based on a human connectivity matrix comprised of 90 different neural populations.

The trade-off between model rank and temporal resolution is also sensitive to prediction delay (Fig 9). For very short prediction delays, increasing the temporal resolution (i.e. decreasing window width) marginally improves the prediction error. For a short time delay, decreasing the window width by a factor of 100 decreased the prediction error by less than 1% for both low and high coupling simulations (Fig 9A and 9C). Thus, increasing the temporal resolution cannot overcome the error penalty incurred by reducing the rank of the approximation. However, as the prediction time delay increases, the improvement due to the reduction in window size (increase in number of windows and consequently system matrices) becomes apparent (Fig 9B and 9D). For a long time delay, decreasing the window width by a factor of 100 decreased the prediction error by about 22% and 16% for the low and high coupling simulations, respectively.

The simulations at low coupling strength exhibited larger errors and smaller $R^2$ values relative to the high coupling strength condition due to greater variance (Fig 9A and 9B). It is worth noting that there is a more marked change in the trends in RMSE and $R^2$ for the high coupling strength case with increasing prediction time delay, despite this condition exhibiting a lower number of attractors and with lower variance among the attractors' fixed point values. This can be attributed to the dynamics of the high coupling case being faster than those of the low coupling case. The faster dynamics are reflected in eigenvalues of larger magnitude, and errors in estimating larger magnitude eigenvalues result in worse predictions.

The timescale that determines what is a short and long prediction delay is given by the eigenvalues of the Jacobian, with the mean eigenvalue for the low coupling strength simulation being around 4 Hz and for the high coupling strength around 27 Hz. The product of the longest prediction time delay (0.15 s) considered with these eigenvalues yields values of about 0.6 and 4, respectively. This disparity in the eigenvalues of the Jacobians between the low and high coupling strength simulations helps explain the different convergence rates of the mean squared error, with the asymptotic behavior we expect based on Eqs 4 and 15 (Fig 7C and 7D). With high coupling strength, there is a stronger dependence on temporal resolution and less dependence on rank as the delay is longer relative to the timescale of the dynamics.

## Clustering of high-dimensional multistable dynamics

As we consider high-dimensional simulations with an increasing number of attractors, the recovery of the attractor landscape becomes more challenging for two main reasons. First, the temporal modes are higher dimensional, thus requiring longer time series for accurate clustering. Second, fixed point attractors' coordinates may differ only in one dimension, thus appearing close in terms of their vector norm. Nonetheless, we observe that the clustering for human connectivity-based simulations and the clustering for the two population simulations follow a similar qualitative trend with increasing prediction delays (Fig 10).

We observe that, for a given distance separation cutoff among clusters, increasing the prediction delay increases the number of temporal mode clusters (Fig 10). For low coupling strength, $G = 0.08$, simulations of the number of clusters increased with increasing delay, whether total variation (TV) regularization (Fig 10A) or spline regularization was used (Fig

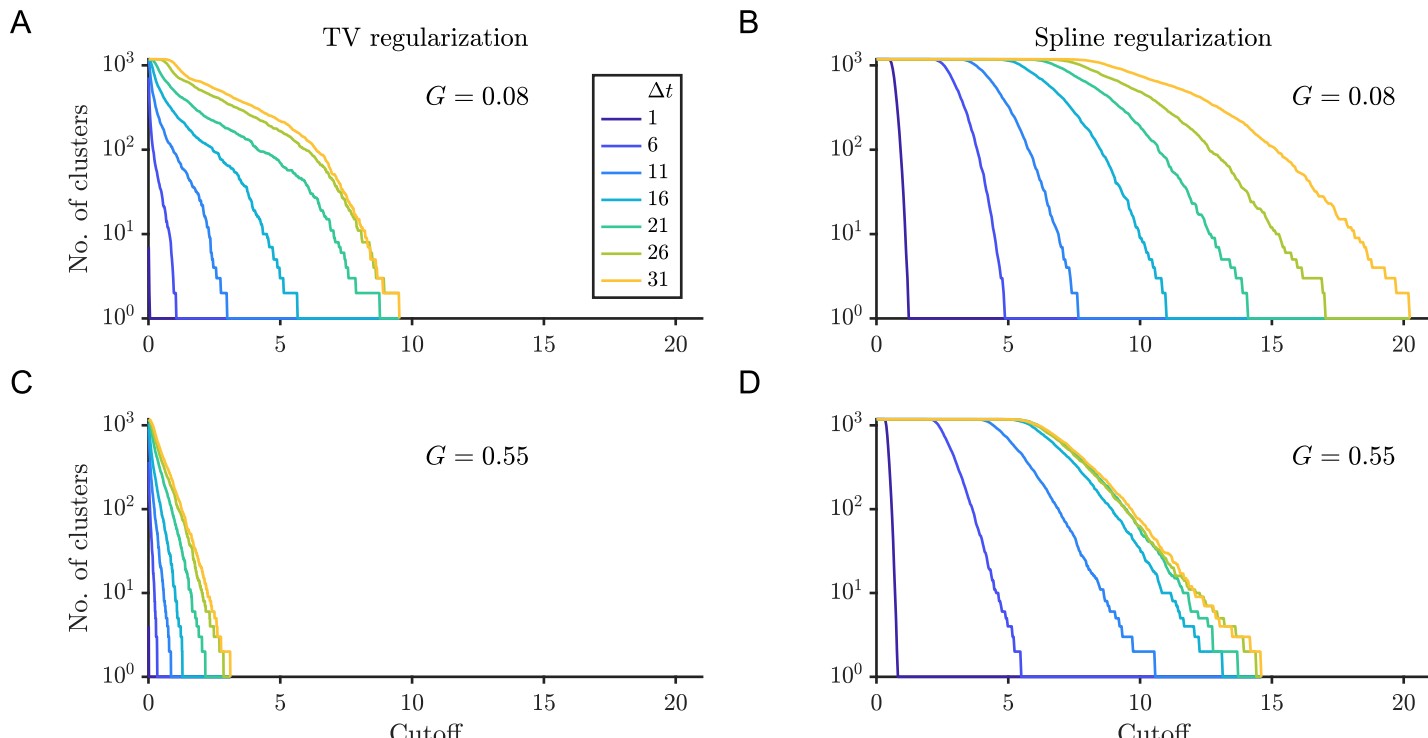

**Fig 10. Hierarchical clustering reveals convergence of number of clusters with prediction delay.** Clustering results for simulations with G = 0.08 (13 equilibria) (A,B) and with G = 0.55 (8 equilibria) (C,D). The number of clusters identified as a function of distance cutoff for several different delay regressions was done with two regularization schemes: total variation (A,C) and spline (B,D). The ground truth number of clusters is given by the number of fixed point attractors.

10B). However, the choice of regularization did impact the overall number of clusters found, as well as the convergence of the clustering curves. TV regularization results in fewer clusters and clustering curves that converge at a delay value around 26 time steps (Fig 10A). No convergence for delays of up to 31 time steps is observed for the curves corresponding to spline regularization (Fig 10B). In the case of the high coupling strength dynamics, the number of clusters for a given cutoff distance converges for delays of 26 or more steps for both the TV (Fig 10C) and spline (Fig 10D) regularizations. The faster convergence of the clustering curves as a function of delay could be due to the faster dynamics of the high coupling case. Consistent with Eq 6 and Fig 7, faster dynamics result in faster convergence of the MSE.

The trend of an increasing number of clusters for increasing prediction delays is observed independent of the regularization scheme used for TVART (Fig 10). These results indicate that the regularization method can impact the identified clusters, with the total variation regularization resulting in fewer clusters overall. However, the clustering for longer prediction delays differs more markedly between simulations with different coupling strengths than the clustering for short delay predictions, irrespective of regularization method (Fig 10). An appropriate choice of time delay can be guided by the convergence of hierarchical clustering, indicating a balance between error and separation between system matrices.

The results presented in this study underscore the importance of accounting for the impact of the time delay between observations when fitting autoregressive models of neural activity. In a manner analogous to attractor reconstruction through time delay embedding, for the identification of time-varying dynamics, an appropriate choice of time delay is a crucial part of the analysis. Our results indicate that the choice of time delay impacts all other metrics, and thus should be optimized along with the other model parameters.

When dealing with multistable systems, such as those arising from neural population dynamics, it is crucial to choose an optimal time delay that allows for the separability of the system matrices while not increasing too much noise. We suggest that for the case of TVART, optimizing parameters should start with fitting full-rank models with short time windows for the data at a range of delays. The convergence of temporal mode clusters (e.g. Fig 10) along with the MSE curve (e.g. Fig 7) provide good indicators for an appropriate time delay. Subsequently, the time window width, $M$, and model rank can be selected to minimize error (e.g. Fig 9).

## Discussion

Identification of different dynamical regimes with potentially similar timescales can be obscured by artifacts that arise from either too fine or too coarse sampling. Conceptually, the autoregressive fitting of stochastic dynamics lies between two extremes. Over infinitesimal time delays, the autoregressive matrix approaches the identity and the error approaches zero. On the other hand, for infinitely long delays the autoregressive matrix approaches zero and the prediction error approaches the variance of the data ($R^2 = 0$). As the results in the previous sections indicate, a single stationary model fits a whole time-varying time series for finely sampled simulations.

For the low-dimensional two population simulations, clustering of TVART temporal mode components is able to recover the attractor dynamics with high accuracy even in the presence of high noise. Moreover, increasing the prediction delay increased the classification accuracy by creating better separated clusters of temporal modes. In the case of simulations based on human brain connectivity, an analogous trend holds. Namely, for a given cutoff distance in hierarchical clustering, a larger delay prediction results in a larger number of clusters, and the effect saturates for the longest prediction horizons.

The choice of $\Delta t$ affects all parameters of the autoregressive process. Perhaps, critically, it also affects the conclusions one may draw from the time-varying autoregressive process analysis. For a very short prediction delay, the dynamics are full rank and time-invariant based on the mean squared error and the $R^2$ value for all simulated scenarios (Fig 9). As the prediction delay increases, it becomes evident that increasing the temporal resolution by reducing the fitting window width can reduce the error in some cases to a larger extent than an increase in the rank approximation (Fig 9). Following the reasoning in the preceding paragraphs, in the limit as $\Delta t \to 0$, the system matrix becomes the full rank identity matrix; conversely, as $\Delta t \to \infty$, the system matrix approaches the zero-rank null matrix. In the latter case, a time-varying affine model may still capture the values of the different fixed points if they are stable enough.

For high-dimensional neural data where the timescale of the dynamics is unknown, fitting curves to Eq (6), could provide an estimate of said timescale. Preliminary analysis of the timescales of the dynamics can help guide the choice of a time delay for prediction for a parsimonious model that considers prediction based on a single observation from the past. Being cognizant of this timescale allows for a selection of time delays that can enhance the identification of separate dynamic regimes. As indicated by the results presented here, choosing an appropriate time delay can maximize the identification of the attractor structure in time series.

Alternatively, autoregression can be performed based on the multistep prediction error. In general, autoregressive methods, including TVART, can easily accommodate methods that include multiple time lags, although this increases the number of parameters and makes the classification problem harder. Iterating a solution over multiple steps can also be used as a strategy for predicting over longer time horizons; such an approach avoids increasing the number of parameters [26]. Nonetheless, the iteration of a linear solution may not be optimal if the underlying dynamics are non-linear. This challenge may be addressed by a novel training method named LINOCS (Lookahead Inference of Networked Operators for Continuous Stability) developed by Mudrik et al. [27]. LINOCS utilizes multiple lookahead estimations to improve the stability and accuracy of dynamical system inference. Future work could explore the combination of LINOCS and TVART in the recovery of attractor dynamics in multistable neural systems.

Moreover, inference by looking further into the future has been shown to be more broadly beneficial. In the case of chaotic neural dynamics, time delay embedding and piecewise linear recurrent neural networks have been leveraged for faithful reconstruction of dynamics [28]. Such an approach in combination with generalized teacher forcing was leveraged by Hess et al. [28] for learning chaotic dynamics. Similarly, RNN-transducers have been shown to benefit from using lookahead tokens, for instance by reducing error in speech to text conversion tasks [29]. These studies, along with the results presented here, underscore the importance of considering the forecast horizon when inferring interpretable neural dynamics. A combination of approaches will probably be required to deal with the full complexity of full brain dynamics, which may include the existence of multiple chaotic attractors [30] as well as a plethora of external stimuli.

In the context of neural time series data, the challenges of high-dimensional nonlinear dynamics, stochasticity, and noise are also compounded by the need to have interpretable dynamics. The results presented here indicate that the locally linear dynamics that can capture the temporal evolution of neural activity accurately can also reveal a broader structure given a parsimonious representation of said dynamics. Such a parsimonious representation can be in terms of the temporal modes of TVART or coefficients from dLDS. In any case, the judicious use of these methods requires more than a goodness-of-fit approach. Just like in the case of attractor reconstruction through time delay embedding, a choice of an appropriate time delay can help extract relevant dynamic features in time-varying auto-regressive models. Moreover,

analyzing error scaling as a function of prediction delay can also reveal global time scales inherent to the dynamics.

One important limitation of the present work is the assumption that observations capture the full state of the system. Another limitation is the assumption of a uniform window length. Moreover, we did not consider the impact of external inputs on the system or the possibility that the underlying nonlinear dynamical system is non-stationary. The TVART model can be extended to address the first three limitations [9, 10], whereas non-stationarity could be inferred after the classification of the system matrices. Alternative tensor-based methods have addressed some of the shortcomings of TVART. For instance, the work by Zhang et al. [31] offers an approach to automatically determine the rank of the tensor decomposition. Also, recent work by Lee et al. [32] offers a novel approach for fitting multiple low rank tensor models, which may be specially useful if the rank if the dynamics varies in the course of a time series (an scenario not considered by TVART).

Although our study mostly focused on using TVART, our findings generalize to other methods of time-varying autoregression (e.g. SLDS, [11], rSLDS [13], and dLDS [16]) insofar as they reveal the importance of balancing dynamic distinguishability with error minimization. Here we demonstrated how both for TVART and dLDS there is an optimal time lag for regression that maximizes attractor identification. Alternative approaches that assume the presence of a finite set of recurring system matrices [11, 14] may be better poised to identify transitions between different dynamics. However, our approach directly explores the matter of whether recurring matrices characterize the dynamics of a time series and, if so, to what extent. We note that both approaches are not mutually exclusive and, once the presence of recurrent dynamics has been identified, it may be beneficial to refine the TVART models with other more computationally expensive methods.

The analytic and numerical results presented in this study indicate that, rather than minimizing the prediction error, identifying different dynamics requires fitting models with a time delay that balances the unfolding of the dynamics with stochastic forcing. System matrices are then able to be grouped into clusters that correspond to distinct attractors of an underlying non-linear system. Thus, it is possible to extract recurrent dynamics in an unsupervised manner by mapping the transitions between clusters over time.

In this study, we know the ground truth dynamics and have complete control over the sampling frequency. This is rarely the case for experimental studies of brain activity, where ground truth dynamics are unknown and there are limitations on measurement resolution and precision. In practice, brain recordings can vary several orders of magnitude in terms of both their temporal and spatial resolutions [33]. In that context, our approach can be applied to elucidate whether the dynamics are under-resolved relative to the time-scale of dynamics (as quantified by the eigenvalues) by exploring prediction error convergence. Thus, sparse identification of dynamics can determine whether a time-varying linear dynamical system model is appropriate for a given data set. Our method also provides an unsupervised technique for obtaining different dynamical regimes while capturing the differences and timing between these regimes. In this manner, it provides a robust tool to identify and characterize recurrent dynamics from neural recordings. Our approach has two major advantages over switching linear dynamics models: it has minimal assumptions about the observed dynamics, and it can find the number and occurrence of recurrent linear systems in an unsupervised manner. Additionally, our approach can test the validity of locally linear assumptions in time-series data by looking at error scaling and inter-cluster distances. Consequently, we believe our method significantly enhances the characterization of dynamic patterns in neural time series, without introducing biases from the experimenter's labeling of the data or from strong assumptions inherent to the model.

## Conclusion

Intuitive, interpretable, and parsimonious characterization of complex dynamics enacted by interacting neural populations remains an elusive goal. In this contribution, we present results that suggest that time-varying autoregression with low rank tensors is a promising tool for identifying different dynamical regimes in the presence of multistable dynamics. This is especially useful when used alongside judicious consideration of the prediction time delay and the inherent timescales of the dynamics. Methods that leverage parsimonious tensor representations of the dynamics jointly with multistep prediction optimization may yield improved insights as to the presence of different dynamical regimes. Moreover, robust methods are crucial for high-dimensional time-varying dynamical systems with multiple dynamical regimes that are subject to strong stochastic forcing. This work shows the crucial role that the timescales of dynamics, noise processes, and attractor switching play in identifying distinct dynamical regimes of neural activity.

## Methods

### Neural mass model simulation details

We summarize the meaning and values of the constants appearing in Eq 1 in Table 1. For the two population simulations, the coupling strengths were 0.32 and 0.03, for the two and the four attractor cases, respectively. For both cases the connectivity is given by $C = \begin{bmatrix} 0 & 1 \\ 1 & 0 \end{bmatrix}$.

The trajectories from the SDEs are generated using the second order stochastic finite difference Heun scheme [34]. The integration time step was 0.1 ms ($\tau_s/1000$) and for analysis the resulting trajectories were down-sampled to a 2 ms and 5 ms sampling period for the low and high-dimensional simulations, respectively. Down-sampling drastically reduced the computational cost of the ensuing analysis and was done with a period faster than that of the fastest dynamics, which are in the order of 100 ms (the timescale set by $\tau_s$). The two population dynamics were simulated for a length of 120 seconds and the 90 population dynamics were simulated for a length of 600 seconds (the first ten seconds of the simulation were dropped to eliminate initial transients).

For human connectivity, the fixed-point attractors are identified by numerical integration of Eq 1 without the additive noise process starting from the initial conditions specified by Latin hypercube sampling of the phase space until convergence of the state variable **S**. Convergence was considered to be achieved when the derivative was two orders of magnitude smaller than the integration time step. After convergence, to better approach the fixed point value, the converged value was passed as an initial estimate to MATLAB's "fsolve" function using a central finite difference estimator.

### Time varying autoregression with low rank tensors

Time-varying autoregression is a widely used time series analysis technique with many applications and implementations [35, 36]. In this work, we consider the TVART technique [9]. The TVART method fits affine models to a sequence of nonoverlapping segments of a time series while imposing a rank constraint to the third order tensor formed by the stack of system matrices via the rank R canonical polyadic decomposition. The rank constraint is a free parameter of TVART and it remains possible to opt for a full rank tensor. Two significant advantages of the TVART approach is that it scales well to problems with many state variables (as is the case for whole brain dynamics) and that it offers a low-dimensional representation of time-

varying dynamics. Thus, TVART may be an ideal framework for identifying distinct dynamical regimes in high-dimensional dynamical systems.

For a time-series of $N$ variables (represented as state vector $\mathbf{x} \in \mathbb{R}^N$) segmented into $T$ windows we aim to find a unique dynamic model for each time window:

$$\mathbf{x}(t + 1) = \mathbf{A}_k \mathbf{x}(t) + \mathbf{b}_k, \tag{16}$$

where the matrix $\mathbf{A}_k$ is the system matrix for the $k^{th}$ time window. TVART uses the rank $R$ canonical polyadic decomposition [17] to represent the tensor formed by the stack of system matrices. Thus, the system matrix fit to window $k$ can be expressed as:

$$\mathbf{A}_k = \mathbf{U}^{(1)} \mathbf{D}^{(k)} \mathbf{U}^{(2)^T}, \tag{17}$$

where $\mathbf{D}^{(k)} = \text{diag}(u_k^{(3)})$, and $u_k^{(3)}$ is the $k^{th}$ row of the temporal mode matrix $U^{(3)} \in \mathbb{R}^{T \times R}$; and $U^{(1)}, U^{(2)} \in \mathbb{R}^{N \times R}$ are called the left and right spatial modes. The spatial modes are constant for the whole time series and the number of non-zero modes can serve as an indication of the true rank of the underlying dynamics [9]. Additionally, the inner product of the rows of the left and right spatial modes determines the overall scale of the interaction of two dynamic variables (for our application the interaction of two neural populations). The system matrix for a given window is uniquely specified by its spatial and temporal mode vectors. This enables a low-dimensional comparison of the dynamics; whereby temporal modes can be clustered to identify similar recurring dynamics.

We used three sets of parameters to apply TVART to our synthetic data sets. For the two population simulations we used parameters $\eta = 0.5$, $\beta = 0.5$, spline regularization, and a window size of 100 observations. For the two population simulations, we used two sets of parameters depending on the regularization scheme. For spline regularization, we used $\eta = 10$, $\beta = 0.1$ and a variable window size as indicated in the results. For total variation (TV) regularization, we used $\eta = 90$, $\beta = 1$ and a window size of 100 observations. Spline regularization was used since the underlying non-linear dynamics are smoothly varying without sharp breaks. The choice of hyperparameters was informed by previous work [9], selecting values for problems of similar size although with relatively weak temporal smoothing regularization.

In order to reduce the computational cost, we did not perform extensive hyperparameter tuning for each TVART fit. However, focusing on two test cases revealed that there is no strong sensitivity to the values of the regularization parameters $\eta$ and $\beta$. We recall the regularized cost function, $C$, used in TVART [9]:

$$C = \frac{1}{2} \sum_{k=1}^{T} \| \mathbf{Y}_k - \mathbf{A}_k \mathbf{X}_k \|_F^2 + \frac{1}{2\eta} \left( \| \mathbf{U}^{(1)} \|_F^2 + \| \mathbf{U}^{(2)} \|_F^2 + \| \mathbf{U}^{(3)} \|_F^2 \right) + \beta \, \mathcal{R}(\mathbf{U}^{(3)}) \tag{18}$$

where the $F$ subscript denotes the Frobenius norm, and the $\mathcal{R}()$ operator denotes either a total variation or a spline penalty used for temporal smoothing [9]. Stronger regularization can result from decreasing $\eta$ (i.e. increasing $1/\eta$) or from increasing $\beta$.

For a parameter sensitivity study we focused on the simulations shown in Figs 3B and 4B with TVART fits with 31 step ahead prediction (the same as used for Figs 3D and 4D). For both cases, we sampled twenty logarithmically spaced values of $\eta$ and $\beta$ spanning four orders of magnitude (between $10^{-2}$ and $10^2$). Moreover, for each hyperparameter pair we fit TVART ten times.

We focus on six metrics to describe the sensitivity of TVART fits to hyperparameter variation. We use $R^2$ as a simple measure of prediction goodness of fit. We consider the log-likelihood of the Gaussian Mixture Model (GMM), in particular we computed the difference in the

log-likelihood values of models with one and either two or four components:

$$\Delta L_n = \text{loglikelihood}(n \text{ components}) - \text{loglikelihood}(1 \text{ component}) \tag{19}$$

we used this to quantify whether a GMM with the same number of components as the number of underlying attractors was more likely than a model with a single Gaussian component. We also compute the average posterior probability of a data point belonging to its assigned cluster, as well as the classification accuracy based on attractor labels. We add two common measures of clustering quality. The Davies-Bouldin index, which is based on the ratio of distances within clusters and between clusters [37], yields smaller values for better clustering. Finally, we computed the Dunn index which is the ratio of shortest distance between observations not in the same cluster to the highest intra-cluster distance [38].

We present the results of this sensitivity study for the two attractor simulations in Fig 11. Although we spanned four orders of magnitude in both hyperparameters, the $R^2$ value barely changed, from 0.81 to 0.89 (Fig 11A). In the case of the log-likelihood, only for the strongest Tikhonov regularization (highest $1/\eta$) did the 1 component GM model perform better than the two component model (Fig 11B). The trends in the mean posterior probability and classification accuracy also showed very robust for a wide range of hyperparameter values (Fig 11C and 11D). Similarly, both the Davies-Bouldin and Dunn indices varied over a small range. As an illustration, we show the clustering results for hyperparameter values very close to those used for the main analyses ($\eta = \beta = 0.48$, Fig 11G), as well as for much larger hyperparameter

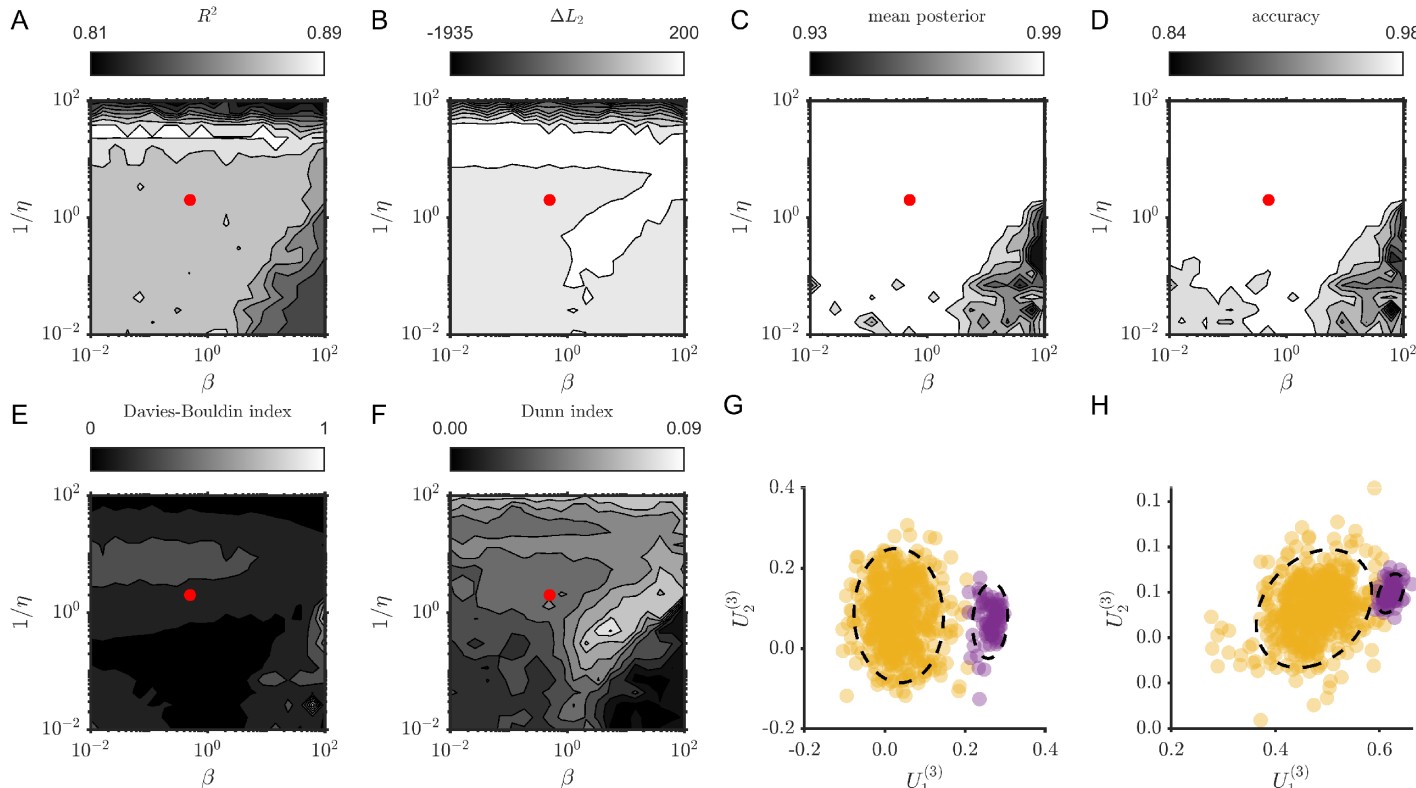

**Fig 11. TVART results are robust over a broad range of hyperparameter values.** (A-F) Contours of the explored metrics as functions of hyperparameters $\eta$ and $\beta$ (increasing $1/\eta$ or $\beta$ leads to stronger regularization), for two-attractor dynamics. The red dot indicate the values used for the main analysis. (G) Clustering example for $\eta = \beta = 0.48$. (H) Clustering example for $\eta = \beta = 100$. The dashed ellipses are the two sigma contours of the Gaussian components.

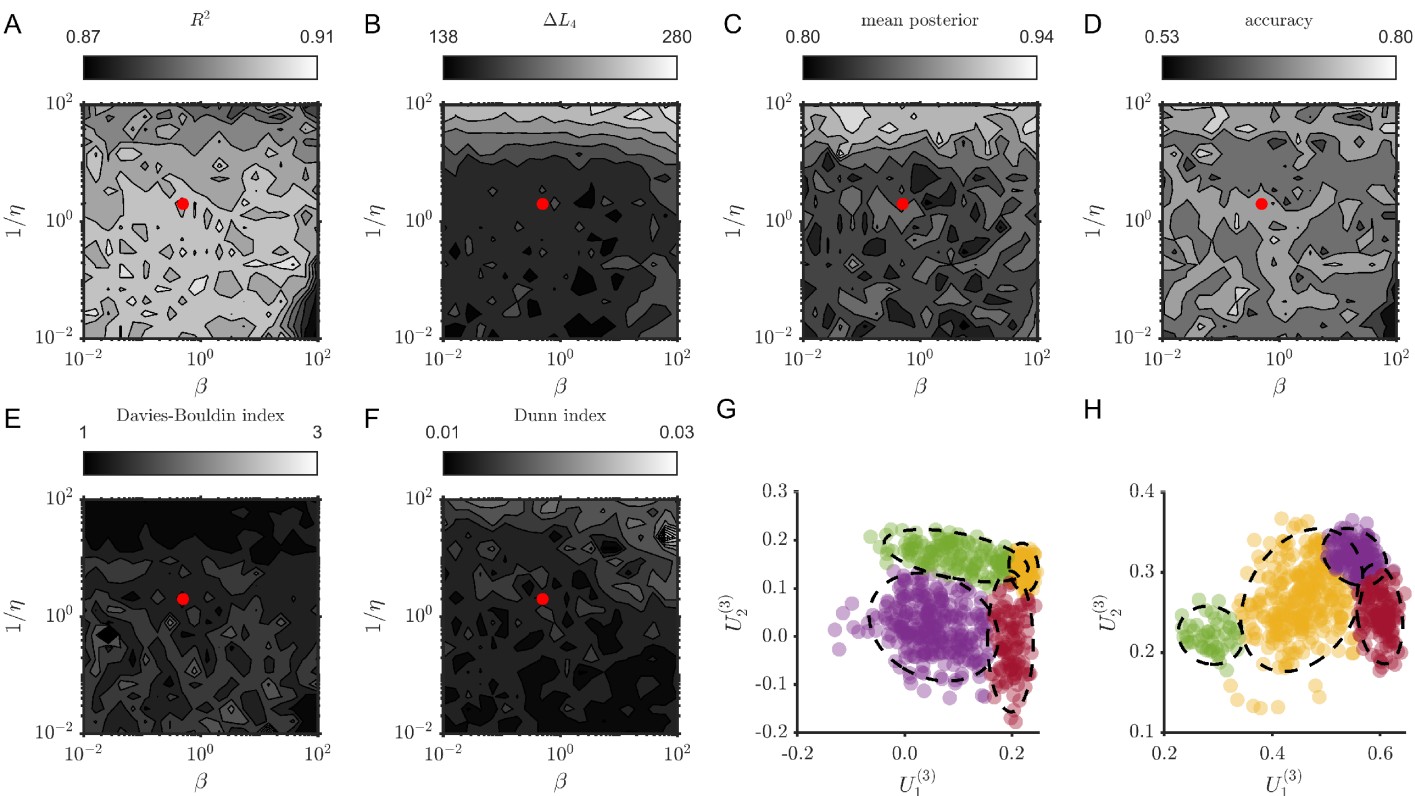

**Fig 12. Greater dynamic complexity only slightly increases hyperparameter sensitivity.** (A-F) Contours of the explored metrics as functions of hyperparameters $\eta$ and $\beta$ (increasing $1/\eta$ or $\beta$ leads to stronger regularization), for four-attractor dynamics. The red dot indicate the values used for the main analysis. (G) Clustering example for $\eta = \beta = 0.48$. (H) Clustering example for $\eta = \beta = 100$. The dashed ellipses are the two sigma contours of the Gaussian components.

values ($\eta = \beta = 100$, Fig 11H). Although clustering is clearly better for $\eta = \beta = 0.48$, at much higher hyperparameter values clustering was still possible.

Results for simulations with four attractors were only slightly more sensitive to hyperparameter variation, as shown in Fig 12. $R^2$ varied merely between 0.87 and 0.91 (Fig 12A). For all the hyperparameter values the log-likelihood of the four component model was greater than that of the one component model (Fig 12B). Both the mean posterior probability and the classification accuracy were more sensitive to hyperparameter variation than they were for the two attractor case (Fig 11C and 11D). This can be expected given the larger number of clusters in this case. Nevertheless, for the mean posterior probability, the range is still small and close to one. And the classification accuracy only showed large gradients for high values of *eta* and $\beta$ (near the lower right corner of Fig 12D). The Davies-Bouldin and Dunn indices only varied by a factor of three, even though the hyperparameters spanned four orders of magnitude (Fig 11E and 11F). The illustrative clustering examples in Fig 11G and 11H, suggest that assigning observations to a cluster was clearly more challenging in this scenario, which may account for the increased sensitivity to hyperparameter variation.

## Clustering

We perform clustering on the temporal mode coefficients via Gaussian Mixture Models [39] for the two population simulations. Gaussian mixture models provide a good metric to evaluate the number of clusters in the data by computing the posterior probability that a point

belongs to a given cluster. However, these models do not scale well for a larger number of clusters in high-dimensional data, especially if the clusters are not well-separated. Thus, we used hierarchical clustering [40] for the high-dimensional simulations based on a human connectivity matrix. The hierarchical clustering is based on the Manhattan distance computed between all possible pairs of the 90-dimensional temporal mode vectors. Hierarchical clustering can inform the number of clusters based on a distance threshold. An optimal number of clusters may be inferred if there is a discontinuity in the curve of number of clusters versus distance threshold. Manhattan distance was considered appropriate since the TVART modes do not constitute an orthonormal basis [9].

## Author Contributions

**Conceptualization:** Rodrigo Osuna-Orozco, Kameron Decker Harris, Samantha R. Santacruz.

**Formal analysis:** Rodrigo Osuna-Orozco.

**Investigation:** Rodrigo Osuna-Orozco.

**Methodology:** Rodrigo Osuna-Orozco, Edward Castillo, Kameron Decker Harris, Samantha R. Santacruz.

**Resources:** Samantha R. Santacruz.

**Writing – original draft:** Rodrigo Osuna-Orozco, Edward Castillo, Kameron Decker Harris, Samantha R. Santacruz.

**Writing – review & editing:** Rodrigo Osuna-Orozco, Edward Castillo, Kameron Decker Harris, Samantha R. Santacruz.

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
