## [Decision Letter · Decision Letter 0]

13 Aug 2024

Dear Osuna-Orozco,

Thank you very much for submitting your manuscript "Identification of Recurrent Dynamics in Distributed Neural Populations" for consideration at PLOS Computational Biology.

As with all papers reviewed by the journal, your manuscript was reviewed by members of the editorial board and by several independent reviewers. In light of the reviews (below this email), we would like to invite the resubmission of a significantly-revised version that takes into account the reviewers' comments.

The reviewers agreed on the methodological relevance of the study and its potential for neuroscience; they also praised the level of detail of the analyses on a range of dynamical regimes.

Reviewers have questions about the breadth of the experimental validation and request further comparison with competing approaches in a neuroscientific context.

Along this line, they expect further clarification on the relationship with neural data and, more broadly, its potential for providing biological insights.

In addition, reviewers home in on the need to clarify concrete methodological aspects. The reviews are constructive and specific, pease find them below. The third review is in a PDF file; please let the editorial staff know if you have difficulty accessing this information.

We cannot make any decision about publication until we have seen the revised manuscript and your response to the reviewers' comments. Your revised manuscript is also likely to be sent to reviewers for further evaluation.

Sincerely,

Emili Balaguer-Ballester, PhD

Academic Editor

PLOS Computational Biology

Daniele Marinazzo

Section Editor

PLOS Computational Biology

Dear authors,

My apologies for the delay in the decision about your manuscript.

The reviewers agreed on the methodological relevance of the study and its potential for neuroscience; they also praised the level of detail of the analyses on a range of dynamical regimes.

Reviewers have questions about the breadth of the experimental validation and request further comparison with competing approaches in a neuroscientific context.

Along this line, they expect further clarification on the relationship with neural data and, more broadly, its potential for providing biological insights.

In addition, reviewers home in on the need to clarify concrete methodological aspects.

We cannot promise publication at this stage, but the reviews are constructive and specific. Please find the reviews below. The third review is in a PDF file; please let the editorial staff know if you have difficulty accessing this information.

Kindly address these queries and submit an updated version if you wish to. Thank you very much.

Best wishes,

Emili Balaguer-Ballester, PhD

Reviewer's Responses to Questions

**Comments to the Authors:**

Reviewer #1: Review for Identification of Recurrent Dynamics in Distributed Neural Populations

Overview: This work explores the TVART dynamical systems model through a series of empirical simulations based on neuroscience models and neural data analysis. TVART, previously presented by Harris et al. (2021), models nonstationary dynamics as switching between a set of dynamics that have a low-rank tensor representation when stacked in a single 3D array. This representation enables a concise description of the dynamics with the assumptions of 1) low-rank dynamics over each window, and 2) locally static linear dynamics that switch between time windows. The primary contributions of this work is a set of experiments that explore the properties of TVART in different contexts and over different dynamical regimes. The rank, stability, and attractor dynamics are specifically studied. While this paper does present interesting analyses of the model, there are several significant weaknesses to the study. For one, the actual experiments are limited compared to the statements in the introduction and the conclusions drawn in the discussion. More significant analyses should be run to justify these statements, such as comparisons to other algorithms or studies of how to better quantify/address some of the properties shown. Furthermore, the analyses are largely circumstantial and some actually tend to suggest that the assumptions in TVART might not best match the data analyzed, reinforcing that comparisons to other methods would be valuable. Finally, as the venue is PLoS CompBio, the link with biology can be strengthened. These points and others are listed with more details below, and I think should be addressed before this work can be considered for publication.

Major points:

1. Clarity of the core analysis: The main analysis of the work is the attractor analysis. The introduction is written such that there was an expectation of a broader analysis of the system properties that was not addressed. My recommendation to the authors is to better align the motivation with the main analysis driving the specific conclusions.

2. Connections to Biology: The connections to biology are primarily through simulation of neurally-motivated synthetic data. It would significantly help the paper in terms of the PLoS Comp Bio community to be more explicit on what the conclusions drawn from these simulation results in terms of neural data analysis.

3. Sparsity vs. Low-rank: The TVART assumes low-rank representations of the dynamics, as emphasized in some of their experiments. The abstract, however, mentions a sparse representation, which is significantly different in terms of underlying model assumptions. Both are low dimensional but are completely different mathematically. What do the authors mean here? Is this different from the sparsity assumption mentioned in the discussion with respect to future work? Is this sparsity different from what was done in the dLDS model or the effective 1-hot sparsity in SLDS?

4. Model Clarity: Understanding the TVART model is pretty critical to understanding the results presented. It would improve the paper a lot to have the explicit form of the model (at least the core equations) presented and described earlier for clarity. This would also help mitigate issues with variables being used that are not defined until the methods.

5. Metrics used: Why do the authors focus on attractor convergence and not the full = trajectory. There can be many dynamical systems with similar attractors but very different paths to get there. Trajectory MSE or other metrics might help measure this.

6. Attractor Hopping: On lines 172-177, the authors state:

“Initially, increasing “the time delay of the prediction resulted in bigger differences among the fitted system matrices. Thus, increasing prediction delay results in clusters of the temporal mode vectors that are more clearly separated, allowing for better classification accuracy (Fig. 3H). But as the prediction delay grows too large, the fit for the system matrices degrades, consequently impacting classification accuracy”

Can this observation be due to the system hopping into another well of attraction? Including an additional analysis that narrows down the possible sources of this effect would be very helpful.

7. Limitations: The limitations of TVART vs other methods in light of the simulations is not sufficiently discussed in the context of the experiments. Low rank A’s seem like they might be more of a significant assumption than authors state in the discussion: “it [TVART] has minimal assumptions about the observed dynamics” as opposed to, e.g., SLDS. In the example in Fig 6, however, large jumps in R^2 for rank changes are observed, seemingly indicating that the data might have a mis-match with the low-rank assumption. Moreover, is this statement in the discussion correct?

“Alternative approaches assume the presence of a finite set of recurring system matrices [12, 17, 28] and thus may be better poised to identify transitions between different dynamics.”

From my understanding TVART also has finite dynamics that are one of a set, finite, linear combination of a smaller set of dynamics (i.e., the low-rank assumption).

8. Other Literature & comparisons: There is minimal discussion of other models and no comparisons to the most relevant methods: SLDS/rSLDS and dLDS.

Additional literature the authors should consider is the work done in developing methods to incorporate longer time-delays into learning, [1-4], which includes work that targets locally linear models similar to TVART [4]. These models all make the same basic observation about time-scales of learning of dynamical systems (and more generally beyond one model) and offer a number of solutions.

On lines 411-412, the authors say “Thus, sparse identification of dynamics can determine whether a time-varying linear dynamical system model is appropriate for a given data set.” Is this not what dLDS already does (Mudrik et al. 2024 in the manuscript’s references).

The authors say that TVART can “can find the number and occurrence of recurrent linear systems in an unsupervised manner.” This statement and others are not demonstrated in the experiments. Comparisons to SLDS on the datasets described would enable these statements to be motivated.

Minor points:

- Fig 3G: Why is 0.9 above 1 on the y axis? If the y-axis is probability it might make sense to scale it to [0, 1] for clarity (as with Fig. 4G/H))

- Line 466: approach is that it is scales

- ‘x’ is not well defined in the first experiment

- Line 152: forgot ‘Fig’ in the \\ref

- This sentence might be better rephrased for clarity: “We quantify prediction accuracy by comparing the clustering results to

- indices representing which attractor the trajectories in a given window are closest to.”

- Consider a log-scale in Figures 3H,3G to highlight the maxima

- In Figure 7 - what is the ground truth number of clusters? Did you also run this example on synthetic Data as well?

- Line 397: no need for commas

References:

[1] Hess, F., Monfared, Z., Brenner, M. and Durstewitz, D., 2023, July. Generalized Teacher Forcing for Learning Chaotic Dynamics. International Conference on Machine Learning (pp. 13017-13049). PMLR.

[2] Massimiliano Marcellino, James H Stock, and Mark W Watson. A comparison of direct and iterated multistep ar methods for forecasting macroeconomic time series. 2006. Journal of econometrics, 135(1-2): 499–526.

[3] Vinit S Unni, Ashish Mittal, Preethi Jyothi, and Sunita Sarawagi. Improving rnn-transducers with acoustic lookahead. 2023. arXiv preprint arXiv:2307.05006.

[4] Mudrik, N., Yezerets, E., Chen, Y., Rozell, C. and Charles, A., 2024. LINOCS: Lookahead Inference of Networked Operators for Continuous Stability. arXiv preprint arXiv:2404.18267.

Reviewer #2: This MS presents simulations aimed to validate TVART, an autorregressive technique to fit low-rank linear models A: X(t+delay) = A X(t) to multivariate timeseries X. The authors use a established neural-mass model to produce two synthetic, one 2-dimensional and one high-dimensional, noisy datasets, and show that TVART can infer the correct attractor dynamics with better-than-chance performance. Moreover, they show that the performance of the autorregressive fit depends non-trivially on the parameter using the delay. Intriguingly, the delay that yields the best performance is not the one yielding the lower prediction error of the fitted model. The authors outline a strategy to identify a regime of near-optimal time-delays for problems where the ground truth is unknown.

The positive outcome of TVART's validation is, in my view, not surprising, given that the original paper validates the technique using several synthetic and real data, including neural data. The non-trivial dependency of performance with the delay, on the other hand, seems new and potentially relevant for researchers that are considering using TVART to analyse high-dimensional multivariate data.

The MS is well written and the methods and results are, to the best of my knowledge, formally correct. However, I am not familiar with autoregression or the analysis of multivariate nerual data so my confidence on this review is limited. I also wonder whether the study would be of interest to the relatively broad readership of PLoS CompBio, or would rather fit better in the journal that published the TVART method (SIAM J Appl Dyn Syst).

I have a few minor suggestions, but the authors should feel free to ignore them if they are not useful.

- As someone not familiar with autoregressive models I had to imagine what "prediction delay" refers to when reading the subsection starting in 144. Given its central role in the results and the scope of PLoS CompBio, it might be worth to introduce the model of Eq.~4 a bit earlier (optimally, before it's used) in the Results section.

- Authors state that they "matched temporal mode cluster indices to attractor indices by choosing the permutation that minimises the difference between the two" (line 163). Although I do not anticipate any methodological problem with this approach, I believe this results in a higher chance level than assumed by the authors. The accuracy in a trial would be sampled from a distribution low-truncated at the original chance level (i.e., no sample could result in an accuracy lower than 1/N_attractors); therefore, the mean would be shifted towards higher accuracies, with chance-level accuracy monotonically increasing with the variance of the single-trial accuracy. If the authors agree with my reasoning, they could account for this by either analytical computation of the mean of the truncated distribution, or by simulating the chance-level accuracy by using uniform random sampling to decide on the attractor and then choosing the most favourable permutation for each trial. This could be important to determine if lower recognition accuracies are indeed above chance levels.

- In the same vein, the authors could compute the chance level as a distributions to show up to what extend their above-chance results could have resulted from the chance distributions.

- Language:

161 grammatically correct, but the preposition at the end makes the sentence a bit difficult to read: consider rephrasing

192 revise the comma

303 missing word (probably "one")

Reviewer #3: Please see attached PDF.

**Have the authors made all data and (if applicable) computational code underlying the findings in their manuscript fully available?**

Reviewer #1: Yes

Reviewer #2: Yes

Reviewer #3: None

PLOS authors have the option to publish the peer review history of their article (what does this mean?). If published, this will include your full peer review and any attached files.

Reviewer #1: No

Reviewer #2: **Yes: **Alejandro Tabas

Reviewer #3: No
---

## [Decision Letter · Decision Letter 1]

27 Nov 2024

PCOMPBIOL-D-24-00905R1

Identification of Recurrent Dynamics in Distributed Neural Populations

PLOS Computational Biology

Dear Dr. Osuna-Orozco,

Thank you for submitting your manuscript to PLOS Computational Biology. After careful consideration, we feel that it has merit but does not fully meet PLOS Computational Biology's publication criteria as it currently stands. Therefore, we invite you to submit a revised version of the manuscript that addresses the points raised during the review process.

Please submit your revised manuscript within 30 days Jan 27 2025 11:59PM. If you will need more time than this to complete your revisions, please reply to this message or contact the journal office at ploscompbiol@plos.org. Please include the following items when submitting your revised manuscript:

We look forward to receiving your revised manuscript.

Kind regards,

Emili Balaguer-Ballester, PhD

Academic Editor

PLOS Computational Biology

Daniele Marinazzo

Section Editor

PLOS Computational Biology

Feilim Mac Gabhann

Editor-in-Chief

PLOS Computational Biology

Jason Papin

Editor-in-Chief

PLOS Computational Biology

**Additional Editor Comments (if provided):**

Dear authors,

The reviewers appreciate your detailed and comprehensive modifications and do not suggest any substantial change. However, please note that reviewer #1 is still concerned about the insufficient citation of previous work.

Yours sincerely,

Emili Balaguer-Ballester, PhD.

**Reviewers' comments:**

Reviewer's Responses to Questions

**Comments to the Authors:**

Reviewer #1: I would like to thank the authors for their thorough and extensive additions to their paper. I believe that the new manuscript, with the additional baselines and clearer discussion, is much improved. However, there are some points that are important and insufficiently addressed. In particular, in the prior comments I had mentioned both additional models (which were in fact included and discussed further by the authors) but also other training methods that address the same challenge/aspect of the work that the authors make one of their main points: the benefit of predicting farther into the future.

Prediction farther into the future (termed look-ahead in prior works) has been studied in multiple models, including in LDS, SLDS, and dLDS models similar to TVART, as well as more general nonlinear systems (RNNs). These need to be discussed as they present significant prior art. I am putting these references here again [1-4]

I also regret that I was not precise about the log-scale for Panels G and H: I meant the y-axis should be on a log scale, not the x-axis. This will emphasize the location of the peak by highlighting subtle height differences into the upper range of values.

References:

- Hess, F., Monfared, Z., Brenner, M. and Durstewitz, D., 2023, July. Generalized Teacher Forcing for Learning Chaotic Dynamics. International Conference on Machine Learning (pp. 13017-13049). PMLR.

- Massimiliano Marcellino, James H Stock, and Mark W Watson. A comparison of direct and iterated multistep ar methods for forecasting macroeconomic time series. 2006. Journal of econometrics, 135(1-2): 499–526.

- Vinit S Unni, Ashish Mittal, Preethi Jyothi, and Sunita Sarawagi. Improving rnn-transducers with acoustic lookahead. 2023. arXiv preprint arXiv:2307.05006.

- Mudrik, N., Yezerets, E., Chen, Y., Rozell, C. and Charles, A., 2024. LINOCS: Lookahead Inference of Networked Operators for Continuous Stability. Transactions of Machine Learning Research 2024.

Reviewer #2: The authors have satisfactorily replied to all my comments, many thanks for your work.

Two notes:

1) I am still not convinced this work will be of true relevance for the audience of PLoS CompBio, but I believe this is a decision that needs to be made by the editor; from my side I can only say that I do not perceive any methodological issues.

2) This is not my field of expertise, so please interpret my endorsement as conditional on the endorsement of other reviewers that do work in this field. I might have overseen potential methodological issues.

Reviewer #3: Thank you for the thoughtful revisions to the paper in response to the comments provided. In particular,

• I appreciate the additional discussion of the TVART model and key terminology at the beginning of the relevant subsection. This clarification enhances the paper significantly. However, I believe that lines 166-171 could benefit from further editing to improve sentence clarity.

• I am pleased to see that relevant methods have been cited and compared. The inclusion of additional experiments to compare TVART with dLDS provides valuable context.

• The discussion of hyperparameters and their selection is very helpful. However, I believe there is still an opportunity to address the sensitivity of TVART to hyperparameter selection in more detail. For example, it would be beneficial if the authors could elaborate on the meaning of “less clear clustering” when beta is increased. A quantitative assessment of clustering quality (and conclusions thereof) as a function of beta (and other hyperparameters) would be helpful.

• I appreciate the expanded discussion on the impact of local minima in TVART model fits.

All in all, I am grateful for the revisions. I believe that the manuscript will be ready for publication with a few minor updates, as described above.

**Have the authors made all data and (if applicable) computational code underlying the findings in their manuscript fully available?**

Reviewer #1: Yes

Reviewer #2: Yes

Reviewer #3: Yes

PLOS authors have the option to publish the peer review history of their article (what does this mean?). If published, this will include your full peer review and any attached files.

Reviewer #1: No

Reviewer #2: **Yes: **Alejandro Tabas

Reviewer #3: No

**Figure resubmission:**
---

## [Decision Letter · Decision Letter 2]

22 Jan 2025

Dear Osuna-Orozco,

We are pleased to inform you that your manuscript 'Identification of Recurrent Dynamics in Distributed Neural Populations' has been provisionally accepted for publication in PLOS Computational Biology.

Best regards,

Emili Balaguer-Ballester, PhD

Academic Editor

PLOS Computational Biology

Daniele Marinazzo

Section Editor

PLOS Computational Biology

Dear authors,

Warm congratulations. Kindly remove the bold font from the Figure 7 caption text when you get the proof. Thank you very much.

Reviewer's Responses to Questions

**Comments to the Authors:**

Reviewer #1: I appreciate the authors work in revising the manuscript. I think the added paragraphs and revised figure are much improved and I'm happy to recommend accepting this manuscript.

Reviewer #3: We thank the authors for addressing all of our comments and concerns. We are happy to recommend the paper for publication!

**Have the authors made all data and (if applicable) computational code underlying the findings in their manuscript fully available?**

Reviewer #1: None

Reviewer #3: Yes

PLOS authors have the option to publish the peer review history of their article (what does this mean?). If published, this will include your full peer review and any attached files.

Reviewer #1: No

Reviewer #3: No

---

## [Editor Report · Acceptance letter]

31 Jan 2025

PCOMPBIOL-D-24-00905R2 

Identification of Recurrent Dynamics in Distributed Neural Populations

Dear Dr Osuna-Orozco,

I am pleased to inform you that your manuscript has been formally accepted for publication in PLOS Computational Biology. Your manuscript is now with our production department and you will be notified of the publication date in due course.

With kind regards,

Zsofia Freund
